# Triple-Optimistic Learning for Stochastic Contextual Bandits with General Constraints

**Hengquan Guo** [1]   **Lingkai Zu** [1]   **Xin Liu** [1]

## Abstract

We study contextual bandits with general constraints, where a learner observes contexts and aims to maximize cumulative rewards while satisfying a wide range of general constraints. We introduce the Optimistic[3] framework, a novel learning and decision-making approach that integrates optimistic design into parameter learning, primal decision, and dual violation adaptation (i.e., triple-optimism), combined with an efficient primal-dual architecture. Optimistic[3] achieves $\tilde{O}(\sqrt{T})$ regret and constraint violation for contextual bandits with general constraints. This framework not only outperforms the state-of-the-art results that achieve $\tilde{O}(T^{\frac{3}{4}})$ guarantees when Slater's condition does not hold but also improves on previous results that achieve $\tilde{O}(\sqrt{T}/\delta)$ when Slater's condition holds ($\delta$ denotes the Slater's condition parameter), offering a $O(1/\delta)$ improvement. Note this improvement is significant because $\delta$ can be arbitrarily small when constraints are particularly challenging. Moreover, we show that Optimistic[3] can be extended to classical multi-armed bandits with both stochastic and adversarial constraints, recovering the best-of-both-worlds guarantee established in the state-of-the-art works, but with significantly less computational overhead.

## 1. Introduction

We study contextual bandits with general constraints, a broad online learning framework that boosts a wide range of applications (e.g., online recommendation (Balakrishnan et al., 2018; Yang et al., 2020), resource-constrained healthcare (Tewari & Murphy, 2017; Tomkins et al., 2021)). In

each round $t$, the learner observes a context $x_t \in \mathcal{X}$, selects an action $a_t \in \mathcal{A}$, and then receives the corresponding noisy reward $r_t(x_t, a_t)$ and cost $c_t(x_t, a_t)$. The goal is to maximize the cumulative reward while ensuring the constraints are satisfied. Our framework addresses a general setting where the cost functions belong to general function classes and can take both positive and negative values. This flexibility allows the model to represent a wide variety of constraints, such as knapsack constraints (Badanidiyuru et al., 2014; 2018; Agrawal & Devanur, 2016; Immorlica et al., 2022), where the cost is positive and the interaction stops once the resource is depleted, replenishable Knapsacks constraints (Kumar & Kleinberg, 2022; Bernasconi et al., 2024a;b), where resources may restore; fairness constraints (Li et al., 2019; Chen et al., 2020; Claure et al., 2020; Sinha, 2024), ensuring equitable outcomes of the decision.

The most closely related literature on contextual bandits with general constraints is (Slivkins et al., 2023; Guo & Liu, 2024). In (Slivkins et al., 2023), the authors proposed the LagrangeCBwLC algorithm to establish the first optimal $\tilde{O}(\sqrt{T}/\delta)$ bounds for both regret and constraint violation. Their results rely on two key assumptions: (1) the existence of Slater's condition, which guarantees a strictly feasible solution, and (2) prior knowledge of Slater's constant $\delta$, which quantifies the strictness of the constraints. These results were later refined by (Guo & Liu, 2024), who achieved $\tilde{O}(T^{\frac{3}{4}})$ regret and violation bounds without requiring Slater's condition. When Slater's condition holds, their $\delta$-agnostic algorithm achieves $\tilde{O}(\sqrt{T}/\delta^2)$ regret and violation, exhibiting a $1/\delta$ worse dependency compared to the full-knowledge algorithm in (Slivkins et al., 2023). The elimination of prior $\delta$-knowledge represents a significant advancement, as obtaining such information is often impractical in real-world applications. However, the $\tilde{O}(T^{\frac{3}{4}})$ regret and violation bounds in (Guo & Liu, 2024) without Slater's condition remain substantially weaker than the optimal $\tilde{O}(\sqrt{T}/\delta)$ bound. For the setting where constraints are tight (i.e., $\delta \to 0$), even the $\tilde{O}(\sqrt{T}/\delta)$ guarantee becomes *problematic*, incurring a critical limitation both in theory and practice. A recent work by (Bernasconi et al., 2024a) shows that $\tilde{O}(\sqrt{T})$ regret and violation bounds are achievable for classical multi-armed bandits with constraints without requiring Slater's condition. However, their approach cannot

---

[1] School of Information Science and Technology, ShanghaiTech University, Shanghai, China. Correspondence to: Xin Liu <liuxin7@shanghaitech.edu.cn>.

*Proceedings of the 42nd International Conference on Machine Learning*, Vancouver, Canada. PMLR 267, 2025. Copyright 2025 by the author(s).

be extended to contextual bandits with general constraints and relies on computationally intensive safe decision set construction. This raises a natural and open question:

*Is there an efficient algorithm for contextual bandits with general constraints that can achieve the optimal $\tilde{O}(\sqrt{T})$ regret and violation **without** the Slater's condition?*

In this paper, we provide a positive answer to this question, and make the following technical contributions: First, we introduce `Optimistic`[3], a framework that incorporates a triple-optimistic design: optimistic estimates, optimistic primal decisions, and optimistic dual updates. The key to these improvements lies in our optimistic dual design. The optimistic dual update is able to more precisely and smoothly predict the expected violation, thus preventing overly aggressive decisions. The optimism in the dual design makes `Optimistic`[3] more adaptive to the constraints, enabling it to timely switch to conservative decisions when overconsumed costs are detected. Second, we establish the first optimal $\tilde{O}(\sqrt{T})$ regret and violation bound in the absence of Slater's condition. This not only significantly improves the state-of-the-art $\tilde{O}(T^{\frac{3}{4}})$ guarantee, but also surpasses the previous best $\tilde{O}(\sqrt{T}/\delta)$ bound with a $1/\delta$ improvement. The $\tilde{O}(\sqrt{T}/\delta)$ guarantees require both Slater's condition and its prior knowledge of Slater's constant. A detailed comparison has been shown in Table 1. Our results can be easily applied to specific types of constraints, including knapsack constraints and fairness constraints. Third, we extend our results into multi-armed bandits with both stochastic and adversarial constraints, and prove the best-of-two-worlds guarantee, covering the state-of-the-art results with greater computational efficiency.

## 2. Related Works

Contextual bandits extend the multi-armed bandit (MAB) framework by incorporating contextual information available at the time of decision-making. In (Dudik et al., 2011; Agarwal et al., 2014), contextual bandits with classification oracles were explored, where the algorithms assume access to cost-sensitive classification oracles. To enhance computational efficiency, contextual bandits with regression oracles were studied in (Foster et al., 2018; Foster & Rakhlin, 2020; Simchi-Levi & Xu, 2022), enabling more computationally efficient designs and making the approach applicable to a wider range of real-world problems, especially in scenarios where accessing classification oracles may not be feasible.

Contextual bandits with constraints by introducing additional costs, requiring the learner to maximize cumulative reward while satisfying constraints. Initial research focused on knapsack constraints (Badanidiyuru et al., 2014; Agrawal & Devanur, 2014; Wu et al., 2015; Agrawal & Devanur, 2016; Badanidiyuru et al., 2018; Sivakumar et al., 2022;

Chzhen et al., 2024; Guo & Liu, 2025), where the interaction terminates upon any constraint violates. For linear function classes, (Agrawal & Devanur, 2016) established the optimal theoretical guarantee of $\tilde{O}((1 + \frac{\nu^*}{b})\sqrt{T})$ was achieved by (Agrawal & Devanur, 2016) for the linear function class. This result was subsequently extended to general function classes by (Han et al., 2023; Slivkins & Foster, 2022) through the introduction of regression oracles for joint reward and cost estimation. A shared assumption across these works is the existence of a null action, which ensures Slater's condition holds.

Fairness constraints have also been extensively studied in the bandit problem. One prominent notion of fairness in multi-armed bandits is the requirement that similar individuals and/or groups are treated similarly, as discussed in (Dwork et al., 2012; Joseph et al., 2016; Chzhen et al., 2024). Another widely explored fairness constraint ensures that each arm receives a minimum fraction of pulls, preventing any arm from being disproportionately neglected (Li et al., 2019; Chen et al., 2020; Claure et al., 2020; Sinha, 2024). There are also studies on group fairness (Chohlas-Wood et al., 2024; Chzhen et al., 2024), where fairness is defined by minimizing the difference in average spending across different groups. These studies design algorithms for specific constraints that incorporate fairness into the decision-making process.

In (Slivkins et al., 2023), a generalization is considered where the constraint should be satisfied in the long term, the cost can be both negative and positive, and the performance of the algorithm is evaluated through both regret and cumulative violation. They provide the first $\tilde{O}(\sqrt{T}/\delta)$ regret and violation guarantee, which, however, relies on the assumption that Slater's condition holds and prior knowledge about it is available. The first theoretical guarantee without Slater's condition is provided by (Guo & Liu, 2024), where they establish $\tilde{O}(T^{\frac{3}{4}})$ regret and violation. When the Slater's condition holds, they also achieve $\tilde{O}(\sqrt{T}/\delta^2)$ regret and violation but do not require prior knowledge of the Slater parameter $\delta$. In the non-context bandit problem, $\tilde{O}(T^{\frac{3}{4}})$ regret and violation is also proved in (Sinha, 2024) without the Slater condition. The optimal $\tilde{O}(\sqrt{T})$ guarantee without Slater's condition in non-context bandits with constraints is provided by (Bernasconi et al., 2024a), however, their algorithm requires constructing a safe decision set for every round, suffering from high computational overhead, especially when the constraints are general and complicated. It remains open whether $\tilde{O}(\sqrt{T})$ regret and violation can be achieved with an efficient algorithm for contextual bandits with general constraints without Slater's condition, as addressed in this paper. A related line of work (Moradipari et al., 2021; Amani et al., 2019; Khezeli & Bitar, 2020; Pacchiano et al., 2024; Gangrade et al., 2024; Chen et al., 2022a; Gangrade & Saligrama, 2025) studies stage-wise

| Reference | Regret/Violation (General constraint) | Regret (Knapsack constraint) | Slater's condition free/ Slater constant agnostic |
|---|---|---|---|
| Optimistic[3] | $\tilde{O}(\sqrt{T})$, $\tilde{O}(\sqrt{T})$ | $O\left((1+\frac{\nu^*}{b})\sqrt{T}\right)$ | ✓, ✓ |
| (Slivkins et al., 2023) | $\tilde{O}(\frac{\sqrt{T}}{\delta})$, $\tilde{O}(\frac{\sqrt{T}}{\delta})$ | $O\left((1+\frac{\nu^*}{b})\sqrt{T}\right)$ | ✗, ✗ |
| (Guo & Liu, 2024) | $\tilde{O}(T^{\frac{3}{4}})$, $\tilde{O}(T^{\frac{3}{4}})$ | × | ✓, ✓ |
| | $\tilde{O}(\frac{\sqrt{T}}{\delta^2})$, $\tilde{O}(\frac{\sqrt{T}}{\delta^2})$ | $O\left((1+\frac{\nu^*}{b^2})\sqrt{T}\right)$ | ✗, ✓ |
| (Chzhen et al., 2024) | × | $O\left((1+\frac{\nu^*}{\delta b})\sqrt{T}\right)$ | ✗, ✗ |
| (Guo & Liu, 2025) | × | $O\left((1+\frac{\nu^*}{\delta b})\sqrt{T}\right)$ | ✗, ✓ |

*Table 1.* Comparison with related works. Optimistic[3] achieves an optimal theoretical guarantee for general constraints *without* requiring the Slater's condition, significantly improving upon the $\tilde{O}(T^{\frac{3}{4}})$ bound in (Guo & Liu, 2024). Even when the Slater's condition holds, our results outperform those in (Guo & Liu, 2024; Slivkins & Foster, 2022) by getting rid of the dependency on $\delta$. Notably, our guarantees for general constraints also recover the state-of-the-art results in (Chzhen et al., 2024; Guo & Liu, 2025), which focus on the knapsack constraints. While (Guo & Liu, 2025) further relaxes the requirement of knowing the Slater's constant assumed in (Chzhen et al., 2024), it still depends on the Slater's condition, whereas our approach does not.

constraints, enforcing safety with high probability at each step. However, these methods rely on an initially known feasible action or the construction of a safe region in every round.

# 3. Stochastic Constrained Contextual Bandits

In this section, we introduce the problem formulation and performance metric for stochastic constrained contextual bandits with general constraints.

**Stochastic Constrained Contextual Bandits:** We study stochastic constrained contextual bandits denoted by $\{\mathcal{X}, \mathcal{A}, \mathcal{F}, \mathcal{G}\}$, where $\mathcal{X}$ is the context set, $\mathcal{A}$ is the action set (a finite set), $\mathcal{F}$ is the reward function class, $\mathcal{G}$ is the cost function class. At period $t$, the learner observes a context $x_t$ randomly generated from the context set $\mathcal{X}$ according to a known probability law $\mathbb{P}(\cdot)$. The learner takes an action $a_t \in \mathcal{A}$, and then receives a random reward $r_t(x_t, a_t) \in [0, 1]$ and a random cost $c_t(x_t, a_t) \in [-1, 1]$. Note that we consider a single constraint for ease of exposition, and our results are readily generalized to multiple constraints. We study a stochastic environment where the arrival of contexts is i.i.d; the observations for rewards and costs are drawn from unknown i.i.d. distributions. We further assume a key general realizability condition for the reward and cost functions (Slivkins et al., 2023; Han et al., 2023; Guo & Liu, 2024; Chzhen et al., 2024).

**Assumption 3.1.** There exists functions $f \in \mathcal{F}$ and $g \in \mathcal{G}$ such that $f(x, a) = \mathbb{E}[r_t(x, a)|x]$ and $g(x, a) = \mathbb{E}[c_t(x, a)|x]$, $\forall x \in \mathcal{X}, a \in \mathcal{A}$.

We define a policy $\pi : \mathcal{X} \to \Delta(\mathcal{A})$, which maps a context to a probability simplex over action set $\mathcal{A}$. Let $\mathbf{f}(x)$ ($\mathbf{r}_t(x)$) and $\mathbf{g}(x)$ ($\mathbf{c}_t(x)$) denote the reward vectors and the cost vectors, respectively, over the action set $\mathcal{A}$ given the context

$x$. The goal of the learner is to design a policy to optimize the cumulative rewards while satisfying the constraints

$$\max_{\pi} \sum_{t=1}^{T} r_t(x_t, a_t) \tag{1}$$

$$\text{s.t.} \sum_{t=1}^{T} c_t(x_t, a_t) \leq 0 \tag{2}$$

The problem formulation in (1)–(2) is general enough to model different types of constraints, including the most common knapsack and fairness constraints:

- (Replenishable) Knapsack constraints (Han et al., 2023; Slivkins et al., 2023; Chzhen et al., 2024; Bernasconi et al., 2024b): Define $c_t(x, a) := w_t(x, a) - b$ and the constraint in (2) represents $\sum_{t=1}^{T} w(x_t, a_t) \leq B$, where $B$ is the initial budget and $b = B/T$ is the average budget (per round). For the most classical bandit with knapsack, the cost function $w_t(x, a)$ is non-negative, and the inter-action would stop once the budget is exhausted. When the knapsack is replenishable, the cost $w_t(x, a)$ could be both positive and negative, where a negative cost implies restoring a positive amount to the budget. The interaction would continue until the end of the time horizon $T$.

- (Min-selection) Fairness constraints (Li et al., 2019; Chen et al., 2020; Claure et al., 2020; Sinha, 2024): Define $c_t(x, a) := \lambda_i - \mathbb{I}\{a = i\}$ for arm $i$ and the constraint in (2) represents $\lambda_i \leq \sum_{t=1}^{T} \mathbb{I}\{a_t = i\}/T$, which means the learner has to choose arm $i$ at least $\lambda_i$ faction of times, i.e., guarantee minimum selection fairness for arm $i$.

- (Group) Fairness constraints (Chohlas-Wood et al., 2024; Chzhen et al., 2024): Define $c_{t,+}(x, a) := w_t(x, a)\mathbb{I}\{\text{gr}(x) = i\} - \gamma_i w_t(x, a)$, and $c_{t,-} :=$

$\gamma_i w_t(x, a) - w_t(x, a)\mathbb{I}\{\text{gr}(x) = i\}$ for group $i$, where $\text{gr}(x) \in [I]$ denotes the group index, and $\gamma_i$ is the expected proportion for group $i$. This constraint reduces disparities in average spending among groups, thereby mitigating equity issues.

**Regret:** To define regret, we first introduce the underlying offline problem of (1)–(2):

$$\max_\pi \; \mathbb{E}\left[\langle \pi, \mathbf{f}(x) \rangle\right] \text{ s.t. } \mathbb{E}\left[\langle \pi, \mathbf{g}(x) \rangle\right] \le 0. \quad (3)$$

Note the optimal value of this offline problem serves as an upper bound to the optimal value of (2), which has been proved in (Devanur et al., 2011; Agrawal & Devanur, 2016; Badanidiyuru et al., 2018). Let $\pi^*$ be the solution to this offline problem and we define $\nu^* := \mathbb{E}\left[\langle \pi^*, \mathbf{f}(x) \rangle\right]$ as the optimal value. For any sequence of action $\{a_t\}_t$, the corresponding (pseudo) regret against this baseline is

$$\mathcal{R}(T) := T\nu^* - \mathbb{E}\left[\sum_{t=1}^T r_t(x_t, a_t)\right]. \quad (4)$$

**Constraint Violation:** The constraint violation is straightforward to be defined as

$$\mathcal{V}(T) := \mathbb{E}\left[\sum_{t=1}^T c_t(x_t, a_t)\right]. \quad (5)$$

Note in the classical contextual bandit with knapsacks, $\mathcal{V}(T) \equiv 0$ due to the "hard stopping" when the budget is exhausted.

**Assumption 3.2.** There exist online learning oracles $\{\mathcal{O}\}_{r,c}$ such that the reward and cost estimators $\hat{f}_t(x, a)$ and $\check{g}_t(x, a)$ satisfy the following conditions with a high probability at least $1 - p$:

$$\mathcal{E} = \begin{cases} 0 \le \hat{f}_t(x, a) - f(x, a) \le 2\varepsilon_t(x, a, p) \\ 0 \le g(x, a) - \check{g}_t(x, a) \le 2\varepsilon_t(x, a, p) \end{cases},$$

where $p = 1/T^2$, and we have $\sum_{t=1}^T \|\varepsilon_t(x, a, p)\|^2 = O(\log(\max(|\mathcal{F}|, |\mathcal{G}|)/p))$.

The above assumption defines two online regression oracles and their performance, which is common in contextual bandits with constraints (Foster et al., 2018; Han et al., 2023; Slivkins et al., 2023; Guo & Liu, 2024; Chzhen et al., 2024; Guo & Liu, 2025). For the (non-context) multi-armed bandits problem, standard UCB (LCB) estimation satisfies the constraint. If the reward and cost functions belong to the generalized linear class, the online least-square estimate oracles satisfy the assumption. When the reward and cost functions are in general class, the weighted online regression estimators still satisfy the assumption, and $\varepsilon_t(x, a, p)$ can be calculated efficiently via a binary search method

(Foster et al., 2018). Based on online learning oracles, we have optimistic learning for rewards and costs.

For completeness, we introduce a key assumption, Slater's condition, used in the existing literature, and note our work does not make this assumption.

**Assumption 3.3** (Slater's condition). There exists a positive constant $\delta$ such that a feasible solution $\pi_0$ to the optimization problem (3) satisfies $\mathbb{E}[\langle \pi_0, \mathbf{g}(x) \rangle] \le -\delta$.

This assumption guarantees the existence of a strictly feasible solution, commonly used in constrained bandit literature (Slivkins et al., 2023; Chzhen et al., 2024; Guo & Liu, 2024; 2025). In contrast, our algorithm eliminates this requirement entirely, and our guarantees are independent of the Slater parameter $\delta$, which may be arbitrarily small in tightly constrained scenarios.

## 4. Triple-optimistic framework for contextual bandits with constraints

In this section, we introduce Optimistic[3] learning and decision framework for contextual bandits with general constraints. This framework incorporates a triple optimistic design in parameter learning, primal decision, and dual violation update to achieve optimal theoretical guarantees.

---

**Optimistic[3] learning and decision framework**

---

**Initialization:** $Q_1 = 0$, $\alpha \ge 2$, uniform policy $\pi_0$, and KL divergence $D(\cdot||\cdot)$.
For $t = 1, \cdots, T$,

- **Optimistic Learning:** Observe the context $x_t$, construct UCB and LCB estimators for rewards $\hat{f}_t(x_t, a)$ and costs $\check{g}_t(x_t, a)$ from the learning oracles $\{\mathcal{O}\}_{r,c}$.

- **Optimistic Decision:** Construct the optimistic surrogate function for any $a \in \mathcal{A}$ that

$$\hat{L}(x_t, a) = \hat{f}_t(x_t, a) - Q_t\check{g}_t(x_t, a). \quad (6)$$

Find the optimal policy $\pi_t$ for probabilistic exploration:

$$\pi_t = \arg\max_\pi \; \langle \pi, \hat{L}(x_t) \rangle - \alpha D(\pi||\pi_{t-1}). \quad (7)$$

Sample an action $a_t \sim \pi_t$.

- **Observe Feedback:** Observe noisy reward $r_t(x_t, a_t)$ and cost $c_t(x_t, a_t)$.

- **Optimistic Dual Update:** Update the optimistic dual variable $Q_{t+1}$ as follows:

$$Q_{t+1} = (Q_t - \mathbb{E}_x[\langle \pi_{t-1}, \check{g}_{t-1}(x) \rangle - 2\langle \pi_t, \check{g}_t(x) \rangle])^+$$

---

$\texttt{Optimistic}^3$ framework is rooted in efficient primal-dual decision-making, leveraging the power of optimism to smoothly control exploration-exploitation tradeoffs under general constraints. It contains the key components:

**Optimistic Parameter Learning:** $\texttt{Optimistic}^3$ incorporates standard learning oracles for reward and cost functions as in (Slivkins et al., 2023; Chzhen et al., 2024; Guo & Liu, 2024). Specifically, the algorithm leverages online regression oracles that collect historical observations of the noisy reward $r_t(x_t, a_t)$ and cost $c_t(x_t, a_t)$. Using these collected samples, the algorithm constructs optimistic estimates of the reward and cost for the next round based on the given context, which is achieved through the UCB estimator for rewards and the LCB estimator for costs.

**Optimistic Primal Decision/Exploration:** Upon the optimistic estimators for reward and cost, the framework first constructs an optimistic surrogate function $\hat{L}(x_t, a)$. We then introduce KL divergence as a regularization term to translate the surrogate function into action probability, which ensures probabilistic, smooth and efficient exploration. This design differs from the "greedy" decision in (Chzhen et al., 2024; Guo & Liu, 2025) and inverse-gap weighting exploration strategy in (Slivkins et al., 2023; Guo & Liu, 2024). Besides, the decision in (7) is very efficient because it is based on online mirror descent and has a closed-form Exp3 update, i.e., $\pi_t(a) = \pi_{t-1}(a)e^{\frac{1}{\alpha}\hat{L}_t(x_t,a)}/\sum_{a'} \pi_{t-1}(a')e^{\frac{1}{\alpha}\hat{L}_t(x_t,a')}$.

**Optimistic Violation Adaptation:** Once the policy $\pi_t$ has been determined, $\texttt{Optimistic}^3$ then conduct optimistic dual update on $Q_t$, which is the key design to provide improved guarantee. Unlike the traditional dual updates

$$Q_{t+1} = (Q_t + \langle \pi_t, \check{\mathbf{g}}_t(x_t) \rangle)^+, \text{ or } (Q_t + c_t(x_t, a_t))^+,$$

which can be interpreted as a standard gradient ascent step on the dual variable, we introduce an additional correction term: $\mathbb{E}_x[\langle \pi_t, \check{\mathbf{g}}_t(x) \rangle] - \mathbb{E}_x[\langle \pi_{t-1}, \check{\mathbf{g}}_{t-1}(x) \rangle]$, which serves as the "momentum" to smooth the update and accelerate convergence. The expectation $\mathbb{E}_x$ denotes expectation over the context variable (drawn from the known context distribution $\mathbb{P}(\cdot)$). The optimistic dual gradient ascent mitigates fluctuations in the dual variable and smooths out abrupt changes in constraint violations across rounds. Interestingly, this optimistic design enables $\texttt{Optimistic}^3$ to achieve the minimal violation even without Slater's condition. The current update requires knowledge of context distribution. When the context distribution is unknown, one can first estimate its empirical distribution and use it in the update. This would not degrade the theoretical analysis when the context distribution satisfies a certain smooth property, as suggested in (Li & Stoltz, 2022; Chen et al., 2022b).

In summary, $\texttt{Optimistic}^3$ presents a novel framework for algorithm design and theoretical analysis in constrained contextual bandit problems. The optimistic and smooth design in primal decisions and dual updates serve as the cornerstone of this framework. By synergistically integrating optimistic estimators with dual variable updates, the optimistic surrogate decision function achieves an elegant balance between reward maximization and cost constraint satisfaction. This approach not only attains optimal theoretical guarantees without requiring Slater's condition but also demonstrates great generality - achieving the best-of-both-worlds in multi-armed bandit with both stochastic and adversarial constraints (as shown in Section 7).

## 5. Theoretical Results

In this section, we present the theoretical performance of $\texttt{Optimistic}^3$ in contextual bandits with general constraints and sketch the proof. We begin by stating the main results in the following theorem, which offers a positive answer to the open question posed in the introduction.

**Theorem 5.1.** *For contextual bandits with general constraints,* $\texttt{Optimistic}^3$ *achieves the following regret and violation:*

$$\mathcal{R}(T) = \tilde{O}(\sqrt{T}), \ \mathcal{V}(T) = \tilde{O}(\sqrt{T}).$$

*Remark* 5.2. These results demonstrate that $\texttt{Optimistic}^3$ achieves strong regret and violation bounds when Slater's condition does not hold. These results improve upon (Guo & Liu, 2024), which establishes $\tilde{O}(T^{\frac{3}{4}})$ regret and violation and only improves to $\tilde{O}(\sqrt{T}/\delta^2)$ regret and violation when Slater's condition holds. While their results focus on the parameter-agnostic setting, (Slivkins et al., 2023) shows that even with prior knowledge of the Slater's condition parameter, the results can only improve to $\tilde{O}(\sqrt{T}/\delta)$ regret and violation, still retaining a dependency on $\delta$. In contrast, $\texttt{Optimistic}^3$ does not require Slater's condition and achieves a clean and elegant $\tilde{O}(\sqrt{T})$ guarantee, eliminating the dependency on $\delta$, which could be small or even vanish in many real-world applications. These results can be further extended into knapsack constraints. Our results significantly advance the theoretical guarantees for contextual bandits with general constraints.

### 5.1. A Unified Bound of "Regret plus Drift"

To prove our main results, we first introduce the following lemma that provides a unified bound for both regret and drift, serving as a key step in the proof of Theorem 5.1. We first define the historical information $\mathcal{H}_t = \{x_t, \hat{f}_t, \check{g}_t, Q_t\}$. The expectation in the following lemma denotes the conditional expectation $\mathbb{E}[\cdot|\mathcal{H}_t]$ given $\mathcal{H}_t$.

**Lemma 5.3.** *Let* $\pi_t$ *be the policy returned by* (7) *in* $\texttt{Optimistic}^3$. *Define a shift dual variable* $q_t = Q_t - \mathbb{E}_x[\langle \pi_{t-1}, \check{\mathbf{g}}_{t-1}(x) \rangle]$ *and its corresponding drift* $\Delta_t =$

$\frac{1}{2}(q_{t+1}^2 - q_t^2)$. *We have*

$$\mathbb{E}[\langle \pi^* - \pi_t, \mathbf{f}(x_t) \rangle] + \mathbb{E}[\Delta_t]$$

$$\leq 2\mathbb{E}[\|\mathbf{f}(x_t) - \hat{\mathbf{f}}_t(x_t)\|] + \mathbb{E}\left[\sum_{t=1}^{T} Q_t \langle \pi^*, \check{\mathbf{g}}_t(x_t) - \mathbf{g}(x_t) \rangle \right]$$

$$+ \alpha\mathbb{E}[D(\pi^*||\pi_{t-1}) - D(\pi^*||\pi_t)]$$

$$+ \mathbb{E}[\|\check{\mathbf{g}}_{t-1}(x) - \check{\mathbf{g}}_t(x)\|^2]$$

$$+ \frac{1}{2}(\mathbb{E}[\langle \pi_t, \check{\mathbf{g}}_t(x) \rangle]^2 - \mathbb{E}[\langle \pi_{t-1}, \check{\mathbf{g}}_{t-1}(x) \rangle]^2)$$

*Remark* 5.4. The above lemma provides a unified bound for both the single-step regret $\langle \pi^* - \pi_t, \mathbf{f}(x_t) \rangle$ and the single-step drift $\Delta_t$, enabling us to analyze regret and violation simultaneously. The drift captures the stability of the dual variable, thereby reflecting the extent to which the total constraint is violated. This lemma refines the results of (Guo & Liu, 2024) by leveraging the power of optimism, allowing us to eliminate additional trade-off parameters between reward and cost that could affect the efficiency of bounding the unified terms and necessitate Slater's condition.

We provide a proof sketch below, with the detailed proof available in Appendix A.

**Proof Sketch:** To begin, we first introduce the following pushback lemma.

**Lemma 5.5** (Pushback Property). *Let $\Pi$ be a convex set. Let function $h$ be convex on $\Pi$ and $\pi_{opt} \in \Pi$ be a global minimum of $h(\pi) + \alpha D(\pi||\pi_{t-1})$ on $\Pi$. For any $\pi \in \Pi$,*

$$h(\pi_{opt}) + \alpha D(\pi_{opt}||\pi_{t-1})$$
$$\leq h(\pi) + \alpha D(\pi||\pi_{t-1}) - \alpha D(\pi||\pi_{opt})$$

Pushback lemma is a fundamental result in online optimization (Nemirovski et al., 2009; Yu et al., 2017; Wei et al., 2020), particularly when working with Bregman divergences (here we specify as KL divergence). This lemma originates from the strong convexity property and guarantees that the optimization process inherently resists deviations from the optimal solution, with the term $\alpha D(\pi||\pi_{opt})$ quantifying the magnitude of this "pushback" effect. By leveraging this property, we can effectively analyze the optimistic decision-making process in (7).

Let $h(\pi) = \langle \pi_{t-1} - \pi, \hat{\mathbf{f}}_t(x_t) \rangle + Q_t \langle \pi, \check{\mathbf{g}}_t(x_t) \rangle$, according to the optimistic decision in (7), we have $\pi_{opt} = \pi_t$. Therefore, we can apply Lemma 5.5 to get that for any policy $\pi \in \Pi$,

$$\langle \pi_{t-1} - \pi_t, \hat{\mathbf{f}}_t(x_t) \rangle + Q_t \langle \pi_t, \check{\mathbf{g}}_t(x_t) \rangle + \alpha D(\pi_t||\pi_{t-1})$$
$$\leq \langle \pi_{t-1} - \pi, \hat{\mathbf{f}}_t(x_t) \rangle + Q_t \langle \pi, \check{\mathbf{g}}_t(x_t) \rangle + \alpha D(\pi||\pi_{t-1})$$
$$- \alpha D(\pi||\pi_t)$$

Let $\pi = \pi^*$ and add $\langle \pi^* - \pi_{t-1}, \mathbf{f}(x_t) \rangle$ on both sides, we

can obtain

$$\langle \pi^* - \pi_t, \mathbf{f}(x_t) \rangle + Q_t \langle \pi_t, \check{\mathbf{g}}_t(x_t) \rangle + \alpha D(\pi_t||\pi_{t-1})$$
$$\leq \langle \pi^* - \pi_t, \mathbf{f}(x_t) - \hat{\mathbf{f}}_t(x_t) \rangle + Q_t \langle \pi^*, \check{\mathbf{g}}_t(x_t) - \mathbf{g}(x_t) \rangle$$
$$+ Q_t \langle \pi^*, \mathbf{g}(x_t) \rangle + \alpha D(\pi^*||\pi_{t-1}) - \alpha D(\pi^*||\pi_t). \quad (8)$$

Recall $q_t = Q_t - \mathbb{E}_x[\langle \pi_{t-1}, \check{\mathbf{g}}_{t-1}(x) \rangle]$ and its corresponding drift $\Delta_t$, the following property holds due to the update rule of the optimistic dual

**Lemma 5.6.** *Under the* `Optimistic`[3] *framework, the following inequality holds for the drift term*

$$\mathbb{E}_x[\Delta_t]$$
$$\leq q_t \mathbb{E}_x[\langle \pi_t, \check{\mathbf{g}}_t(x) \rangle] + \mathbb{E}_x[\langle \pi_t, \check{\mathbf{g}}_t(x) \rangle]^2$$
$$= Q_t \mathbb{E}_x[\langle \pi_t, \check{\mathbf{g}}_t(x) \rangle] - \mathbb{E}_x[\langle \pi_{t-1}, \check{\mathbf{g}}_{t-1}(x) \rangle]\mathbb{E}_x[\langle \pi_t, \check{\mathbf{g}}_t(x) \rangle]$$
$$+ \mathbb{E}_x[\langle \pi_t, \check{\mathbf{g}}_t(x) \rangle]^2 \quad (9)$$

The cross term $\mathbb{E}_x[\langle \pi_{t-1}, \check{\mathbf{g}}_{t-1}(x) \rangle]\mathbb{E}_x[\langle \pi_t, \check{\mathbf{g}}_t(x) \rangle]$ is challenging to analyze directly, so we introduce the following property to facilitate the analysis:

**Lemma 5.7.** *The following inequality holds for $t \in [T]$,*

$$\mathbb{E}_x[\langle \pi_{t-1}, \check{\mathbf{g}}_{t-1}(x) \rangle]\mathbb{E}_x[\langle \pi_t, \check{\mathbf{g}}_t(x) \rangle]$$
$$\geq \frac{\mathbb{E}_x[\langle \pi_t, \check{\mathbf{g}}_t(x) \rangle]^2}{2} + \frac{\mathbb{E}_x[\langle \pi_{t-1}, \check{\mathbf{g}}_{t-1}(x) \rangle]^2}{2}$$
$$- \mathbb{E}_x[\|\check{\mathbf{g}}_{t-1}(x) - \check{\mathbf{g}}_t(x)\|^2] - \|\pi_t - \pi_{t-1}\|_1^2 \quad (10)$$

*Remark* 5.8. Suppose the algorithm is designed without leveraging any prior knowledge of the context distribution. In this case, the optimistic dual update takes the form

$$Q_{t+1} = (Q_t - \langle \pi_{t-1}, \check{\mathbf{g}}_{t-1}(x_{t-1}) \rangle + 2\langle \pi_t, \check{\mathbf{g}}_t(x_t) \rangle)^+.$$

As shown in Lemma 5.7, this update introduces the term $\|\check{\mathbf{g}}_{t-1}(x_{t-1}) - \check{\mathbf{g}}_t(x_t)\|^2$, which poses significant analytical challenges, as it reflects the complexity of context-dependent variations. To circumvent this issue, we have to adopt the mild assumption that the context distribution is known and leverage this knowledge in the dual update. An important direction for future work is to develop techniques that remove the assumption of prior knowledge of the context distribution.

With the above two lemmas, we can take expectation w.r.t. $x_t$ and substitute (9) and (10) into (8), which gives

$$\langle \pi^* - \pi_t, \mathbb{E}_x[\mathbf{f}(x_t)] \rangle + \Delta_t$$
$$\leq \langle \pi^* - \pi_t, \mathbb{E}_x[\mathbf{f}(x_t) - \hat{\mathbf{f}}_t(x_t)] \rangle + \mathbb{E}_x[Q_t \langle \pi^*, \check{\mathbf{g}}_t(x_t) - \mathbf{g}(x_t) \rangle]$$
$$+ \alpha D(\pi^*||\pi_{t-1}) - \alpha D(\pi^*||\pi_t) + \mathbb{E}_x[\|\check{\mathbf{g}}_{t-1}(x) - \check{\mathbf{g}}_t(x)\|^2]$$
$$+ \frac{\mathbb{E}_x[\langle \pi_t, \check{\mathbf{g}}_t(x) \rangle]^2 - \mathbb{E}_x[\langle \pi_{t-1}, \check{\mathbf{g}}_{t-1}(x) \rangle]^2}{2}$$
$$+ \|\pi_t - \pi_{t-1}\|_1^2 - \alpha D(\pi_t||\pi_{t-1}) + \mathbb{E}_x[Q_t \langle \pi^*, \mathbf{g}(x_t) \rangle].$$

For the last term, we have $\mathbb{E}_x[Q_t\langle\pi^*, \mathbf{g}(x_t)\rangle] = 0$ according to the definition of $\pi^*$. Then it remains to prove $\|\pi_t - \pi_{t-1}\|_1^2 - \alpha D(\pi_t\|\pi_{t-1}) \leq 0$, for which the following lemma holds:

**Lemma 5.9** (Pinsker's inequality). *Let $\pi_t$ and $\pi_{t-1}$ be two probability distributions over action set $\mathcal{A}$ and $D(\cdot\|\cdot)$ be KL Divergence, we have*

$$\|\pi_{t-1} - \pi_t\|_1^2 \leq 2D(\pi_t\|\pi_{t-1})$$

Applying Pinsker's inequality and recalling $\alpha \geq 2$, then we can take the expectation w.r.t. the remaining terms in $\mathcal{H}_t$ to complete the proof. We might also consider alternative generating functions in the Bregman divergence, which yield properties similar to Pinsker's inequality and obtain the same guarantees as in Theorem 5.1. This might require selecting an appropriate learning rate $\alpha$ in our algorithm such that the key Lemma 5.3 holds.

## 5.2. Regret analysis

To establish the regret bound, we first sum the inequality in Lemma 5.3 to get

$$\mathbb{E}\left[\sum_{t=1}^T\langle\pi^* - \pi_t, \mathbf{f}(x_t)\rangle\right] + \mathbb{E}\left[\sum_{t=1}^T\Delta_t\right]$$

$$\leq\mathbb{E}\left[\sum_{t=1}^T\|\mathbf{f}(x_t) - \hat{\mathbf{f}}_t(x_t)\| + \sum_{t=1}^T Q_t\langle\pi^*, \check{\mathbf{g}}_t(x_t) - \mathbf{g}(x_t)\rangle\right]$$

$$+ \mathbb{E}\left[\sum_{t=1}^T\alpha(D(\pi^*\|\pi_{t-1}) - D(\pi^*\|\pi_t))\right] \tag{11}$$

$$+ \mathbb{E}\left[\sum_{t=1}^T\|\check{\mathbf{g}}_{t-1}(x) - \check{\mathbf{g}}_t(x)\|^2\right] \tag{12}$$

$$+ \sum_{t=1}^T\left(\frac{\mathbb{E}[\langle\pi_t, \check{\mathbf{g}}_t(x)\rangle]^2}{2} - \frac{\mathbb{E}[\langle\pi_{t-1}, \check{\mathbf{g}}_{t-1}(x)\rangle]^2}{2}\right) \tag{13}$$

The first two terms denote the learning errors of reward and cost functions, for which we have the following upper bound due to our learning oracle assumptions.

$$\mathbb{E}\left[\sum_{t=1}^T\|\mathbf{f}(x_t) - \hat{\mathbf{f}}_t(x_t)\|\right] = \tilde{O}(\sqrt{T}),$$

$$\mathbb{E}\left[\sum_{t=1}^T Q_t\langle\pi^*, \check{\mathbf{g}}_t(x_t) - \mathbf{g}(x_t)\rangle\right] = \tilde{O}(1),$$

where the last one holds since we have $Q(t) \leq 2T$ from the update rule of the dual variable.

From the telescoping nature of (11) and (13) and the upper

bounds for each component, we can easily derive:

$$\mathbb{E}\left[\sum_{t=1}^T\alpha(D(\pi^*\|\pi_{t-1}) - D(\pi^*\|\pi_t))\right] = \tilde{O}(1),$$

$$\sum_{t=1}^T\left(\frac{\mathbb{E}[\langle\pi_t, \check{\mathbf{g}}_t(x)\rangle]^2}{2} - \frac{\mathbb{E}[\langle\pi_{t-1}, \check{\mathbf{g}}_{t-1}(x)\rangle]^2}{2}\right) = O(1),$$

Finally, for the terms in (12), we have

$$\mathbb{E}\left[\sum_{t=1}^T\|\check{\mathbf{g}}_{t-1}(x) - \check{\mathbf{g}}_t(x)\|^2\right]$$

$$\leq\mathbb{E}\left[\sum_{t=1}^T\|\check{\mathbf{g}}_{t-1}(x) - \mathbf{g}(x)\|^2\right] + \mathbb{E}\left[\sum_{t=1}^T\|\check{\mathbf{g}}_t(x) - \mathbf{g}(x)\|^2\right]$$

$$\leq 2\mathbb{E}\left[\sum_{t=1}^T\|\varepsilon_t(x, a, p)\|^2\right] = \tilde{O}(1),$$

where the last inequality holds due to the learning oracle assumption. Combining all these terms, we prove the upper bound of regret such that $\mathcal{R}(T) = \tilde{O}(\sqrt{T})$. Please refer to Appendix B.1 for the detailed proof.

## 5.3. Violation analysis

To establish the cumulative violation guarantee, we first need to analyze the relationship between violation and dual variable. According to the update rule of $Q_t$, we have

$$Q_{t+1} \geq Q_t - \mathbb{E}_x[\langle\pi_{t-1}, \check{\mathbf{g}}_{t-1}(x)\rangle] + 2\mathbb{E}_x[\langle\pi_t, \check{\mathbf{g}}_t(x)\rangle].$$

Summing the above inequality over $T$ rounds and taking expectation, we can obtain that:

$$\mathbb{E}\left[\sum_{t=1}^T\langle\pi_t, \check{\mathbf{g}}_t(x)\rangle\right] \leq Q_{T+1} + 1 \leq q_{T+1} + 2.$$

This demonstrates that the expected estimated cumulative violation can be upper bounded by the "dual" term $q_{T+1}$.

Then we can prove the constraint violation through the following decomposition:

$$\mathcal{V}(T) = \mathbb{E}\left[\sum_{t=1}^T\langle\pi_t, \mathbf{g}(x) - \check{\mathbf{g}}_t(x)\rangle\right] + \mathbb{E}\left[\sum_{t=1}^T\langle\pi_t, \check{\mathbf{g}}_t(x)\rangle\right]$$

$$\leq\mathbb{E}\left[\sum_{t=1}^T\|\mathbf{g}(x) - \check{\mathbf{g}}_t(x)\|\right] + q_{T+1} + 2, \tag{14}$$

where we can complete the proof by analyzing the cumulative learning error and the upper bound of $q_T$. Recall in Section 5.2, we have proved a unified bound for regret plus drift, which gives

$$\mathbb{E}\left[\sum_{t=1}^T\langle\pi^* - \pi_t, \mathbf{f}(x_t)\rangle\right] + \mathbb{E}\left[\sum_{t=1}^T\Delta_t\right] = \tilde{O}(\sqrt{T}),$$

substitute the definition of drift, recall $Q_1 = 0$ and rearrange the inequality, we can obtain

$$\mathbb{E}\left[\frac{q_{T+1}^2}{2}\right] = \tilde{O}(\sqrt{T}) + \mathbb{E}\left[\sum_{t=1}^{T}\langle \pi_t - \pi^*, \mathbf{f}(x_t)\rangle\right]$$

$$\leq \tilde{O}(\sqrt{T}) + 2T,$$

where the last inequality holds due to the bound of the reward function. Finally, we show that $q_{T+1} = \tilde{O}(\sqrt{T})$, we then prove the constraint violation by substituting it into (14).

## 6. `Optimistic`[3] for CBwK

This section instantiates `Optimistic`[3] for contextual bandits with knapsack constraints and shows how the theoretical guarantee in Theorem 5.1 can be translated into the knapsack constraints with "hard-stopping". Intuitively, the constraint violation is a proxy for the early stopping and will be accounted for regret.

For the knapsack constraints, we have $c_t(x_t, a_t) = w_t(x_t, a_t) - b$ and $\sum_{t=1}^{T} w_t(x_t, a_t) \leq B$. The regret definition is the same as in (4). To present our results, we first define the stopping time under our algorithm, where $\tau = \arg\min_{t\in[T]}\{t \mid \sum_{s=1}^{t} w(x_t, a_t) \geq B\}$. We then decouple the regret as follows

$$\mathcal{R}(T) = \mathbb{E}\left[\sum_{t=1}^{\tau}(\nu^* - r_t(x_t, a_t))\right] + (T - \tau)\nu^*.$$

The first term represents the "regret before stopping" and can be bounded using our regret analysis of contextual bandits with general constraints, i.e., $\mathcal{R}(T)$. Specifically, it is bounded by $\tilde{O}(\sqrt{\tau}) = \tilde{O}(\sqrt{T})$ in Theorem 5.1.

The second term, $(T - \tau)\nu^*$ represents the "regret after stopping," and its upper bound can be determined by analyzing the upper bound of the constraint violation. From the violation bound in Theorem 5.1, we can immediately have $\mathbb{E}\left[\sum_{t=1}^{\tau}(w_t(x_t, a_t) - b)\right] = \tilde{O}(\sqrt{T})$, in conjunction with the definition of stopping time, we know $\tau$ satisfies $\tilde{O}(\sqrt{T}) + \tau b \geq B := Tb$, which establish that $T - \tau = \tilde{O}(\sqrt{T}/b)$ such that

$$\mathbb{E}[(T - \tau)\nu^*] = \tilde{O}\left(\frac{\nu^*}{b}\sqrt{T}\right).$$

These results are summarized in the following theorem, and its proof can be found in Appendix C.

**Theorem 6.1.** *For contextual bandits with knapsack constraints,* `Optimistic`[3] *achieves the regret that*

$$\mathcal{R}(T) = \tilde{O}\left((1 + \frac{\nu^*}{b})\sqrt{T}\right).$$

The regret bound holds for a universal budget regime where $B = \Omega(1)$ without assuming Slater's condition or the information of Slater's constant. This greatly improves upon the state-of-the-art in (Chzhen et al., 2024; Guo & Liu, 2025), where guarantees are limited to a small-budget regime $B = \Omega(\sqrt{T})$, with (Chzhen et al., 2024) additionally requiring explicit knowledge of Slater's condition.

## 7. `Optimistic`[3] for MAB with Stochastic and Adversarial Constraints

A very recent paper (Bernasconi et al., 2024a) studied classical (non-contextual) multi-armed bandits (MAB) with stochastic and adversarial constraints, which achieve "best-of-two-worlds" guarantees by constructing the feasible set through adaptive weighted estimates. Interestingly, they also establish an optimal $\tilde{O}(\sqrt{T})$ regret guarantee without Slater's condition under the stochastic setting. We also illustrate `Optimistic`[3] can be readily generalized to MAB with both stochastic and adversarial constraints and recover the "best-of-two-worlds" results with a much less computational overhead as `Optimistic`[3] has gradient-type decision while (Bernasconi et al., 2024a) requires solving linear programming for each round.

In MAB with stochastic and adversarial constraints, at time $t$, the learner chooses an arm $a_t \in \mathcal{A}$ and receives a reward $r_t(a_t)$ and a cost $c_t(a_t)$. Similar to (Bernasconi et al., 2024a), we consider the reward to be arbitrary and adversarial. Then the cost functions are the key to dividing this problem.

**Stochastic Setting.** The cost at each round is independently drawn from an i.i.d. distribution with expected value $c(a) = \mathbb{E}[c_t(a)], \forall a \in \mathcal{A}, t \in [T]$. The baseline is defined as the best fixed policy that satisfies the constraints in expectation. Consequently, the definitions of regret and constraint violation remain the same as those in the main text, with the slight modification that the contexts are omitted.

**Adversarial Setting.** The cost sequence $\{c_t\}_t$ can arbitrary. As the baseline, we consider the best unconstrained policy:

$$\pi_{adv}^* := \arg\max_{\pi}\sum_{t=1}^{T}\langle \pi, \mathbf{r}_t\rangle.$$

In this more challenging setting, as a sublinear regret result is impossible (Immorlica et al., 2022; Castiglioni et al., 2022), we aim for a less ambitious objective: achieving a constant competitive ratio with respect to $\text{OPT}_{adv} = \sum_{t=1}^{T}\langle \pi_{adv}^*, \mathbf{r}_t\rangle$, the cumulative reward of the unconstrained optimal policy. Formally, given the competitive ratio $\beta \leq 1$, we establish the following $\beta$-regret as:

$$\beta\text{-}\mathcal{R}(T) = \beta\,\text{OPT}_{adv} - \mathbb{E}\left[\sum_{i=1}^{T} r_t(a_t)\right].$$

The definition of violation remains unchanged.

In the stochastic setting, (Bernasconi et al., 2024a) also achieves the optimal $\tilde{O}(\sqrt{T})$ regret and violation without Slater's condition, a result that is, of course, covered by our work. In the adversarial setting, their analysis further requires Slater's condition to hold, where $\rho := -\inf_{a \in \mathcal{A}} \max_{t \in [T]} c_t(a)$ is the Slater constant. They consider the competitive ratio to be $\beta = \rho/(1 + \rho)$ and achieve $\tilde{O}(\sqrt{T})$ regret and violation. To achieve these "best-of-two-worlds" guarantees, they estimate the cost function using a weighted average of past observations combined with an optimistic bonus. They adaptively design the weights based on the current level of constraint violation, allowing the estimates to accommodate the differing requirements of stochastic and adversarial constraints. However, the algorithm proposed in Bernasconi et al. (2024a) incurs additional computational overhead, as it constructs a feasible set using estimated costs and solves a constrained optimization problem at each round. Next, we will show that `Optimistic`[3] can recover their performance guarantees while retaining a gradient-based update scheme to ensure computational efficiency.

To apply `Optimistic`[3] to this problem, we only need to modify the optimistic learning components for the reward and constraint functions. Specifically, to handle adversarial rewards, we can use standard EXP3-style estimators. To accommodate both stochastic and adversarial constraints, we adopt the adaptive weighted estimator proposed in (Bernasconi et al., 2024a). The details of the `Optimistic`[3] algorithm for MAB with stochastic and adversarial constraints are provided in the Appendix D. For this problem, we have the following guarantee:

**Theorem 7.1.** *For MAB with stochastic and adversarial constraints,* `Optimistic`[3] *achieves the following results for stochastic constraints*

$$\mathcal{R}(T) = \tilde{O}(\sqrt{T}), \ \mathcal{V}(T) = \tilde{O}(\sqrt{T}),$$

*and for adversarial constraints with $\beta = \rho/(1 + \rho)$, under the Slater's condition,* `Optimistic`[3] *achieves*

$$\beta\text{-}\mathcal{R}(T) = \tilde{O}(\sqrt{T}), \ \mathcal{V}(T) = \tilde{O}(\sqrt{T}).$$

The above theorem shows that `Optimistic`[3] achieves optimal $\tilde{O}(\sqrt{T})$ regret and violation in the stochastic setting without assuming Slater's condition. For adversarial constraints, let $\beta = \rho/(1 + \rho)$; under the Slater's condition, `Optimistic`[3] achieves $\tilde{O}(\sqrt{T})$ $\beta$-regret and violation. These results match the "best-of-both-worlds" guarantees established in (Bernasconi et al., 2024a), while `Optimistic`[3] achieves these results with significantly less computational overhead.

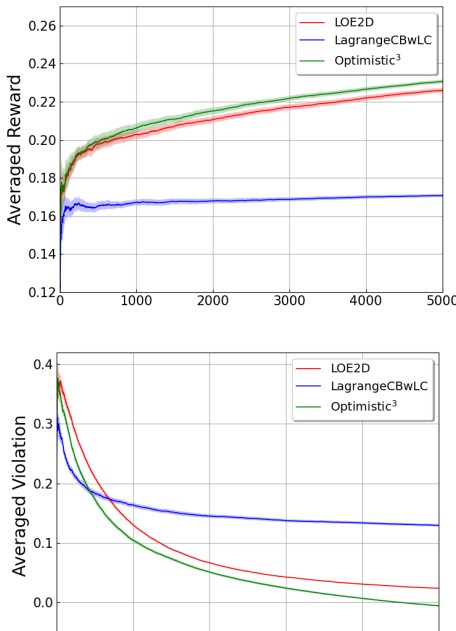

*Figure 1.* Averaged reward and constraint violation under LOE2D, LagrangeCBwLC and `Optimistic`[3].

## 8. Experiments

In this section, we conduct experiments using the large-scale learning-to-rank dataset Microsoft MSLR-WEB30k (Qin & Liu, 2013). We adopt a similar general constraint setting as in (Guo & Liu, 2024), where the reward function $r(x, a)$ corresponds to the relevance score (normalized to $[0, 1]$) assigned to the recommended document and the incoming customer. The cost $c(x, a)$ for each arm is randomly generated from a uniform distribution over $[-0.5, 1]$ and remains fixed throughout each trial. We employ gradient-boosted tree regression and the empirical mean to estimate reward and cost functions, respectively. Observations of rewards and costs are perturbed by Gaussian noise $\mathcal{N}(0, 0.05)$. The reported experimental results are averaged over 50 trials, with a 95% confidence interval. Figure 1 demonstrates that `Optimistic`[3] outperforms both LOE2D (Guo & Liu, 2024) and LagrangeCBwLC (Slivkins et al., 2023), which verifies our theoretical results.

## 9. Conclusion

In this paper, we propose `Optimistic`[3], an efficient and optimal framework for contextual bandits with general constraints. `Optimistic`[3] significantly improves the theoretical guarantees in the setting where Slater's condition does not hold. Furthermore, we demonstrate that these results can be extended to the non-contextual MAB setting, achieving the best-of-both-worlds guarantee for MAB with stochastic and adversarial constraints.

## Acknowledgments

This work was supported by the National Natural Science Foundation of China under Grant 62302305 and the Core Facility Platform of Computer Science and Communication, SIST, ShanghaiTech University.

## Impact Statement

This paper presents work whose goal is to advance the field of Optimization. There are many potential societal consequences of our work, none which we feel must be specifically highlighted here.

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

# A. Proof of the key property in Lemma 5.3

In this section, we provide detailed proof of Lemma 5.3, which provides a unified bound for both regret and violation. We begin by establishing the following pushback property (for a detailed discussion and analysis, we refer the reader to (Wei et al., 2020), which provides the first proof of the general pushback lemma).

**Lemma A.1** (Restatement of Lemma 5.5). *Let $\mathcal{X}$ be a convex set. Let function $h$ be convex on $\Pi$, and let $\pi_{opt} \in \Pi$ be a global minimum of $h(\pi) + D(\pi||\pi_{t-1})$ on $\Pi$. Then, for any $\pi \in \Pi$, we have:*

$$h(\pi_{opt}) + \alpha D(\pi_{opt}||\pi_{t-1}) \leq h(\pi) + \alpha D(\pi||\pi_{t-1}) - \alpha D(\pi||\pi_{opt})$$

*Proof.* Due to the definition of $\pi_{opt}$, we have

$$\nabla h(\pi_{opt}) + \alpha \nabla D(\pi_{opt}||\pi_{t-1}) = 0,$$

A standard "three-point" expansion of the second bracket shows

$$D(\pi||\pi_{t-1}) - D(\pi_{opt}||\pi_{t-1}) = D(\pi||\pi_{opt}) + \langle \nabla D(\pi_{opt}||\pi_{t-1}), \pi - \pi_{opt} \rangle,$$

Recall $\nabla h(\pi_{opt}) + \nabla D(\pi_{opt}||\pi_{t-1}) = 0$, then we can get

$$(h(\pi) - h(\pi_{opt})) + \alpha(D(\pi||\pi_{t-1}) - D(\pi_{opt}||\pi_{t-1})) = (h(\pi) - h(\pi_{opt}) - \langle \nabla h(\pi_{opt}), \pi - \pi_{opt} \rangle) + \alpha D(\pi||\pi_{opt})$$
$$\geq \alpha D(\pi||\pi_{opt}),$$

Rearranging the above inequality, we complete the proof. $\qquad\square$

We consider the case conditioned on the historical information $\mathcal{H}_t$. Then, we can apply the pushback lemma to our decision process in (7), where $h(\pi) = \langle \pi_{t-1} - \pi, \hat{\mathbf{f}}_t(x_t) \rangle + Q_t \langle \pi, \check{\mathbf{g}}_t(x_t) \rangle$, which implies that for any policy $\pi \in \Pi$:

$$\langle \pi_{t-1} - \pi_t, \hat{\mathbf{f}}_t(x_t) \rangle + Q_t \langle \pi_t, \check{\mathbf{g}}_t(x_t) \rangle + \alpha D(\pi_t||\pi_{t-1})$$
$$\leq \langle \pi_{t-1} - \pi, \hat{\mathbf{f}}_t(x_t) \rangle + Q_t \langle \pi, \check{\mathbf{g}}_t(x_t) \rangle + \alpha D(\pi||\pi_{t-1}) - \alpha D(\pi||\pi_t)$$

Let $\pi = \pi^*$ and add $\langle \pi^* - \pi_{t-1}, \mathbf{f}(x_t) \rangle$ on both sides, we can obtain

$$\langle \pi^* - \pi_t, \mathbf{f}(x_t) \rangle + Q_t \langle \pi_t, \check{\mathbf{g}}_t(x_t) \rangle + \alpha D(\pi_t||\pi_{t-1})$$
$$\leq \langle \pi^* - \pi_t, \mathbf{f}(x_t) - \hat{\mathbf{f}}_t(x_t) \rangle + Q_t \langle \pi^*, \check{\mathbf{g}}_t(x_t) - \mathbf{g}(x_t) \rangle + Q_t \langle \pi^*, \mathbf{g}(x_t) \rangle + \alpha D(\pi^*||\pi_{t-1}) - \alpha D(\pi^*||\pi_t),$$

To address the term $Q_t \langle \pi_t, \check{\mathbf{g}}_t(x_t) \rangle$, recall that $q_t = Q_t - \mathbb{E}_x[\langle \pi_{t-1}, \check{\mathbf{g}}_{t-1}(x) \rangle]$, and the corresponding drift is $\Delta_t = \frac{q_{t+1}^2}{2} - \frac{q_t^2}{2}$. For the drift term, the following property holds:

**Lemma A.2** (Restatement of Lemma 5.6). *Under the* `Optimistic`³ *framework, the following inequality holds for the drift term*

$$\mathbb{E}_x[\Delta_t] \leq q_t \mathbb{E}_x[\langle \pi_t, \check{\mathbf{g}}_t(x) \rangle] + \mathbb{E}_x[\langle \pi_t, \check{\mathbf{g}}_t(x) \rangle]^2$$
$$= Q_t \mathbb{E}_x[\langle \pi_t, \check{\mathbf{g}}_t(x) \rangle] - \mathbb{E}_x[\langle \pi_{t-1}, \check{\mathbf{g}}_{t-1}(x) \rangle] \mathbb{E}_x[\langle \pi_t, \check{\mathbf{g}}_t(x) \rangle]$$
$$+ \mathbb{E}_x[\langle \pi_t, \check{\mathbf{g}}_t(x) \rangle]^2$$

*Proof.* Recall the definition of $q_t$ and the optimistic dual update:

$$Q_{t+1} = (Q_t - \mathbb{E}_x[\langle \pi_{t-1}, \check{\mathbf{g}}_{t-1}(x) \rangle] + 2\mathbb{E}_x[\langle \pi_t, \check{\mathbf{g}}_t(x) \rangle])^+.$$

We can derive the update rule of $q_{t+1}$:

$$q_{t+1} + \mathbb{E}_x[\langle \pi_t, \check{\mathbf{g}}_t(x) \rangle] = (q_t + 2\mathbb{E}_x[\langle \pi_t, \check{\mathbf{g}}_t(x) \rangle])^+$$
$$q_{t+1} = \max(-\mathbb{E}_x[\langle \pi_t, \check{\mathbf{g}}_t(x) \rangle], q_t + \mathbb{E}_x[\langle \pi_t, \check{\mathbf{g}}_t(x) \rangle]),$$

To facilitate the analysis of $q_t$, we define the following $\phi(\pi_t)$:

$$\phi(\pi_t) = \begin{cases} \mathbb{E}_x[\langle \pi_t, \check{\mathbf{g}}_t(x) \rangle], & \text{if } q_t + \mathbb{E}_x[\langle \pi_t, \check{\mathbf{g}}_t(x) \rangle] \geq -\mathbb{E}_x[\langle \pi_t, \check{\mathbf{g}}_t(x) \rangle] \\ -q_t - \mathbb{E}_x[\langle \pi_t, \check{\mathbf{g}}_t(x) \rangle], & \text{else.} \end{cases}$$

Then we can rewrite the update of $q_{t+1}$ as

$$q_{t+1} = q_t + \phi(\pi_t).$$

Recall the definition of the drift term, we have

$$\begin{aligned} \Delta_t =& \frac{\phi(\pi_t)^2}{2} + q_t \phi(\pi_t) \\ =& \frac{\phi(\pi_t)^2}{2} + q_t \mathbb{E}_x[\langle \pi_t, \check{\mathbf{g}}_t(x) \rangle] + q_t(\phi(\pi_t) - \mathbb{E}_x[\langle \pi_t, \check{\mathbf{g}}_t(x) \rangle]) \\ =& \frac{\phi(\pi_t)^2}{2} + q_t \mathbb{E}_x[\langle \pi_t, \check{\mathbf{g}}_t(x) \rangle] - (\phi(\pi_t) + \mathbb{E}_x[\langle \pi_t, \check{\mathbf{g}}_t(x) \rangle])(\phi(\pi_t) - \mathbb{E}_x[\langle \pi_t, \check{\mathbf{g}}_t(x) \rangle]) \\ =& q_t \mathbb{E}_x[\langle \pi_t, \check{\mathbf{g}}_t(x) \rangle] + \mathbb{E}_x[\langle \pi_t, \check{\mathbf{g}}_t(x) \rangle]^2 - \frac{\phi(\pi_t)^2}{2} \\ \leq& q_t \mathbb{E}_x[\langle \pi_t, \check{\mathbf{g}}_t(x) \rangle] + \mathbb{E}_x[\langle \pi_t, \check{\mathbf{g}}_t(x) \rangle]^2, \end{aligned}$$

where the third equality follows from the fact that $q_t(\phi(\pi_t) - \mathbb{E}_x[\langle \pi_t, \check{\mathbf{g}}_t(x) \rangle]) = (\phi(\pi_t) + \mathbb{E}_x[\langle \pi_t, \check{\mathbf{g}}_t(x) \rangle])(\phi(\pi_t) - \mathbb{E}_x[\langle \pi_t, \check{\mathbf{g}}_t(x) \rangle])$, which can be proven by considering the cases $\phi(\pi_t) = \mathbb{E}_x[\langle \pi_t, \check{\mathbf{g}}_t(x) \rangle]$ and $\phi(\pi_t) \neq \mathbb{E}_x[\langle \pi_t, \check{\mathbf{g}}_t(x) \rangle]$. Finally, we substitute $q_t = Q_t - \mathbb{E}_x[\langle \pi_{t-1}, \check{\mathbf{g}}_{t-1}(x) \rangle]$, which completes the proof. $\square$

When analyzing the upper bound of the drift term, the cross term $\mathbb{E}_x[\langle \pi_{t-1}, \check{\mathbf{g}}_{t-1}(x) \rangle]\mathbb{E}_x[\langle \pi_t, \check{\mathbf{g}}_t(x) \rangle]$ presents significant analytical challenges. To resolve this, we derive the following lemma.

**Lemma A.3** (Restatement of Lemma 5.7). *The following inequality holds for $t \in [T]$,*

$$\begin{aligned} \mathbb{E}_x[\langle \pi_{t-1}, \check{\mathbf{g}}_{t-1}(x) \rangle]\mathbb{E}_x[\langle \pi_t, \check{\mathbf{g}}_t(x) \rangle] \geq& \frac{\mathbb{E}_x[\langle \pi_t, \check{\mathbf{g}}_t(x) \rangle]^2}{2} + \frac{\mathbb{E}_x[\langle \pi_{t-1}, \check{\mathbf{g}}_{t-1}(x) \rangle]^2}{2} \\ & - \mathbb{E}_x[\|\check{\mathbf{g}}_{t-1}(x) - \check{\mathbf{g}}_t(x)\|^2] - \|\pi_t - \pi_{t-1}\|_1^2 \end{aligned}$$

*Proof.*

$$\begin{aligned} (\mathbb{E}_x[\langle \pi_t, \check{\mathbf{g}}_t(x) \rangle - \langle \pi_{t-1}, \check{\mathbf{g}}_{t-1}(x) \rangle])^2 =& (\mathbb{E}_x[\langle \pi_t, \check{\mathbf{g}}_t(x) - \check{\mathbf{g}}_{t-1}(x) \rangle] + \mathbb{E}_x[\langle \check{\mathbf{g}}_{t-1}, \pi_t - \pi_{t-1} \rangle])^2 \\ \leq& (\mathbb{E}_x[\|\pi_t\| \|\check{\mathbf{g}}_t(x) - \check{\mathbf{g}}_{t-1}(x)\|] + \mathbb{E}_x[\|\check{\mathbf{g}}_{t-1}\|_\infty \|\pi_t - \pi_{t-1}\|_1])^2 \\ \leq& (\mathbb{E}_x[\|\check{\mathbf{g}}_t(x) - \check{\mathbf{g}}_{t-1}(x)\|] + \mathbb{E}_x[\|\check{\mathbf{g}}_{t-1}\|_\infty \|\pi_t - \pi_{t-1}\|_1])^2 \\ \leq& 2\mathbb{E}_x[\|\check{\mathbf{g}}_t(x) - \check{\mathbf{g}}_{t-1}(x)\|^2] + 2\|\pi_t - \pi_{t-1}\|_1^2, \end{aligned}$$

where the first inequality comes from Hölder's inequality, the second and third inequality hold since $\|\pi_t\| \leq 1$ and $\|\check{\mathbf{g}}_{t-1}\|_\infty \leq 1$, the last inequality holds since $(a+b)^2 \leq 2a^2 + 2b^2$ and $\mathbb{E}[X]^2 \leq \mathbb{E}[X^2]$. Then we can obtain

$$\frac{\mathbb{E}_x[\langle \pi_t, \check{\mathbf{g}}_t(x) \rangle]^2}{2} + \frac{\mathbb{E}_x[\langle \pi_{t-1}, \check{\mathbf{g}}_{t-1}(x) \rangle]^2}{2} - \mathbb{E}_x[\langle \pi_{t-1}, \check{\mathbf{g}}_{t-1}(x) \rangle]\mathbb{E}_x[\langle \pi_t, \check{\mathbf{g}}_t(x) \rangle] \leq \mathbb{E}_x[\|\check{\mathbf{g}}_{t-1}(x) - \check{\mathbf{g}}_t(x)\|^2] + \|\pi_t - \pi_{t-1}\|_1^2.$$

Rearranging these terms and then we complete the proof. $\square$

By combining the two lemmas above with the inequality derived from the pushback property and applying Pinsker's inequality——which states $\|\pi_{t-1} - \pi_t\|_1^2 \leq 2D(\pi_t\|\pi_{t-1}) \leq \alpha D(\pi_t\|\pi_{t-1})$——we derive the following bound:

$$\begin{aligned} \mathbb{E}_x[\langle \pi^* - \pi_t, \mathbf{f}(x_t) \rangle | \mathcal{H}_t] + \mathbb{E}_x[\Delta_t | \mathcal{H}_t] \leq& \mathbb{E}_x[2\|\mathbf{f}(x_t) - \hat{\mathbf{f}}_t(x_t)\| | \mathcal{H}_t] + \mathbb{E}_x\left[ \sum_{t=1}^T Q_t \langle \pi^*, \check{\mathbf{g}}_t(x_t) - \mathbf{g}(x_t) \rangle | \mathcal{H}_t \right] \\ & + \mathbb{E}_x[\alpha D(\pi^*\|\pi_{t-1}) - \alpha D(\pi^*\|\pi_t) | \mathcal{H}_t] \\ & + \mathbb{E}_x[\|\check{\mathbf{g}}_{t-1}(x) - \check{\mathbf{g}}_t(x)\|^2 | \mathcal{H}_t] \\ & + \frac{1}{2}(\mathbb{E}[\langle \pi_t, \check{\mathbf{g}}_t(x) \rangle | \mathcal{H}_t]^2 - \mathbb{E}[\langle \pi_{t-1}, \check{\mathbf{g}}_{t-1}(x) \rangle | \mathcal{H}_t]^2), \end{aligned}$$

Taking expectation w.r.t. the historical information $\mathcal{H}_t$, we complete the proof of Lemma 5.3.

# B. Proof of Theorem 5.1

In this section, we prove our main results in Theorem 5.1 by leveraging Lemma 5.3, which establishes a unified bound for both regret and Lyapunov drift. We first derive the regret bound, followed by the constraint violation analysis.

## B.1. Proof of Regret bound

We begin by summing the inequality in Lemma 5.3 over $T$ to obtain:

$$\mathbb{E}\left[\sum_{t=1}^{T}\langle\pi^* - \pi_t, \mathbf{f}(x_t)\rangle\right] + \mathbb{E}\left[\sum_{t=1}^{T}\Delta_t\right] \leq \mathbb{E}\left[2\sum_{t=1}^{T}\|\mathbf{f}(x_t) - \hat{\mathbf{f}}_t(x_t)\|\right]$$

$$+ \mathbb{E}\left[\sum_{t=1}^{T}Q_t\langle\pi^*, \check{\mathbf{g}}_t(x_t) - \mathbf{g}(x_t)\rangle\right]$$

$$+ \mathbb{E}\left[\sum_{t=1}^{T}\alpha(D(\pi^*||\pi_{t-1}) - D(\pi^*||\pi_t))\right]$$

$$+ \sum_{t=1}^{T}\mathbb{E}\left[\|\check{\mathbf{g}}_{t-1}(x) - \check{\mathbf{g}}_t(x)\|^2\right]$$

$$+ \mathbb{E}\left[\sum_{t=1}^{T}\left(\frac{\langle\pi_t, \check{\mathbf{g}}_t(x)\rangle^2}{2} - \frac{\langle\pi_{t-1}, \check{\mathbf{g}}_{t-1}(x)\rangle^2}{2}\right)\right],$$

We analyze these terms one by one. For the first term, based on the learning oracle assumption, we can obtain:

$$\mathbb{E}\left[2\sum_{t=1}^{T}\|\mathbf{f}(x_t) - \hat{\mathbf{f}}_t(x_t)\|\right] \leq 2|\mathcal{A}|\sum_{t=1}^{T}\|\varepsilon_t(x, a, p)\| \leq 2|\mathcal{A}|\sqrt{T\sum_{t=1}^{T}\|\varepsilon_t(x, a, p)\|^2}$$

$$= 2|\mathcal{A}|\sqrt{T\log(\max(|\mathcal{F}|, |\mathcal{G}|)/p)} = \tilde{O}(\sqrt{T}),$$

where the second inequality comes from the Cauchy-Schwarz inequality. For the term $\mathbb{E}\left[\sum_{t=1}^{T}Q_t\langle\pi^*, \check{\mathbf{g}}_t(x_t) - \mathbf{g}(x_t)\rangle\right]$, we decompose the expectation using the event $\mathcal{E}$ defined in Assumption 3, considering both $\mathcal{E}$ and its complement $\bar{\mathcal{E}}$.

$$\mathbb{E}\left[\sum_{t=1}^{T}Q_t\langle\pi^*, \check{\mathbf{g}}_t(x_t) - \mathbf{g}(x_t)\rangle\right] = \mathbb{E}\left[\sum_{t=1}^{T}Q_t\langle\pi^*, \check{\mathbf{g}}_t(x_t) - \mathbf{g}(x_t)\rangle|\mathcal{E}\right] + \mathbb{E}\left[\sum_{t=1}^{T}Q_t\langle\pi^*, \check{\mathbf{g}}_t(x_t) - \mathbf{g}(x_t)\rangle|\bar{\mathcal{E}}\right]$$

$$\leq \mathbb{E}\left[\sum_{t=1}^{T}Q_t\langle\pi^*, \check{\mathbf{g}}_t(x_t) - \mathbf{g}(x_t)\rangle|\bar{\mathcal{E}}\right]$$

$$\leq \mathbb{E}\left[\sum_{t=1}^{T}\|Q_t\|\|\pi^*\|\|\check{\mathbf{g}}_t(x_t) - \mathbf{g}(x_t)\|\|\bar{\mathcal{E}}\right]$$

$$\leq 4|\mathcal{A}|pT^2,$$

where the last inequality holds since the event $\bar{\mathcal{E}}$ holds with probability $p$ and the facts that $\|Q_t\| \leq 2T$, $\|\pi^*\| \leq 1$, $\|\check{\mathbf{g}}_t(x_t) - \mathbf{g}(x_t)\| \leq 2|\mathcal{A}|$. Then we show that $\mathbb{E}\left[\sum_{t=1}^{T}Q_t\langle\pi^*, \check{\mathbf{g}}_t(x_t) - \mathbf{g}(x_t)\rangle\right] = O(1)$ since $p = \frac{1}{T^2}$.

Next we analyze $\mathbb{E}\left[\sum_{t=1}^{T}\alpha(D(\pi^*||\pi_{t-1}) - D(\pi^*||\pi_t))\right]$ and $\mathbb{E}\left[\sum_{t=1}^{T}\left(\frac{\langle\pi_t, \check{\mathbf{g}}_t(x)\rangle^2}{2} - \frac{\langle\pi_{t-1}, \check{\mathbf{g}}_{t-1}(x)\rangle^2}{2}\right)\right]$. For these two terms, it's easy to get that

$$\mathbb{E}\left[\sum_{t=1}^{T}\alpha(D(\pi^*||\pi_{t-1}) - D(\pi^*||\pi_t))\right] = \mathbb{E}[\alpha D(\pi^*||\pi_0)]$$

$$\leq \alpha\log(|\mathcal{A}|),$$

the inequality holds due to the uniform initialization of $\pi_0$.

$$\sum_{t=1}^{T}\left(\frac{\mathbb{E}\left[\langle\pi_t,\check{\mathbf{g}}_t(x)\rangle\right]^2}{2}-\frac{\mathbb{E}\left[\langle\pi_{t-1},\check{\mathbf{g}}_{t-1}(x)\rangle\right]^2}{2}\right)=\frac{\mathbb{E}\left[\langle\pi_T,\check{\mathbf{g}}_T(x)\rangle\right]^2}{2}$$
$$\leq\frac{1}{2},$$

where the last inequality holds since $\check{g}_T(x,a)\in[-1,1]$ and $\pi_T\in\Pi$ is the probability distribution. Finally, we have

$$\mathbb{E}\left[\sum_{t=1}^{T}\|\check{\mathbf{g}}_{t-1}(x)-\check{\mathbf{g}}_t(x)\|^2\right]\leq\mathbb{E}\left[\sum_{t=1}^{T}\|\check{\mathbf{g}}_{t-1}(x)-\mathbf{g}(x)\|^2\right]+\mathbb{E}\left[\sum_{t=1}^{T}\|\check{\mathbf{g}}_t(x)-\mathbf{g}(x)\|^2\right]$$
$$\leq2|\mathcal{A}|\mathbb{E}\left[\sum_{t=1}^{T}\|\varepsilon_t(x,a,p)\|^2\right]$$
$$\leq2|\mathcal{A}|\log(\max(|\mathcal{F}|,|\mathcal{G}|)/p)=\tilde{O}(1),$$

where the last inequality holds due to our learning oracle assumption. Combining all these terms, then for the regret term, we have

$$\mathbb{E}\left[\sum_{t=1}^{T}\langle\pi^*-\pi_t,\mathbf{f}(x_t)\rangle\right]$$
$$\leq-\mathbb{E}\left[\sum_{t=1}^{T}\Delta_t\right]+2|\mathcal{A}|\sqrt{T\log(\max(|\mathcal{F}|,|\mathcal{G}|)/p)}+4|\mathcal{A}|pT^2+\alpha\log(|\mathcal{A}|)+2|\mathcal{A}|\log(\max(|\mathcal{F}|,|\mathcal{G}|)/p)+\frac{1}{2}$$
$$=-\mathbb{E}\left[\frac{q_1^2}{2}\right]+2|\mathcal{A}|\sqrt{T\log(\max(|\mathcal{F}|,|\mathcal{G}|)/p)}+4|\mathcal{A}|pT^2+\alpha\log(|\mathcal{A}|)+2|\mathcal{A}|\log(\max(|\mathcal{F}|,|\mathcal{G}|)/p)+\frac{1}{2}$$
$$\leq2|\mathcal{A}|\sqrt{T\log(\max(|\mathcal{F}|,|\mathcal{G}|)/p)}+4|\mathcal{A}|pT^2+\alpha\log(|\mathcal{A}|)+2|\mathcal{A}|\log(\max(|\mathcal{F}|,|\mathcal{G}|)/p)+\frac{1}{2}=\tilde{O}(\sqrt{T}),$$

where the last inequality holds since the initialization of $Q_1$ and we set $\mathbb{E}_x[\langle\pi_0,\check{\mathbf{g}}_0(x)\rangle]=0$, then we complete the proof of the regret bound.

### B.2. Proof of Violation bound

From the update rule of $Q_t$, we have

$$\mathbb{E}\left[\sum_{t=1}^{T}\langle\pi_t,\check{\mathbf{g}}_t(x)\rangle\right]\leq Q_{T+1}+1\leq q_{T+1}+2.$$

Then the cumulative constraint violation can be bounded by

$$\mathbb{E}\left[\sum_{t=1}^{T}\langle\pi_t,\mathbf{g}(x)\rangle\right]=\mathbb{E}\left[\sum_{t=1}^{T}\langle\pi_t,\mathbf{g}(x)-\check{\mathbf{g}}_t(x)\rangle\right]+\mathbb{E}\left[\sum_{t=1}^{T}\langle\pi_t,\check{\mathbf{g}}_t(x)\rangle\right]$$
$$\leq\mathbb{E}\left[\sum_{t=1}^{T}\|\mathbf{g}(x)-\check{\mathbf{g}}_t(x)\|\right]+q_{T+1}+2,$$

Recall that we have already proved a unified upper bound for both regret and violation, which gives

$$\mathbb{E}\left[\sum_{t=1}^{T}\langle\pi^*-\pi_t,\mathbf{f}(x_t)\rangle\right]+\mathbb{E}\left[\sum_{t=1}^{T}\Delta_t\right]\leq2|\mathcal{A}|\sqrt{T\log(\max(|\mathcal{F}|,|\mathcal{G}|)/p)}+4|\mathcal{A}|pT^2+\alpha\log(|\mathcal{A}|)+\frac{1}{2},$$

Rearrange these terms, we have

$$\mathbb{E}\left[\sum_{t=1}^{T}\Delta_t\right] = \mathbb{E}\left[\frac{q_{T+1}^2}{2}\right] \leq \mathbb{E}\left[\sum_{t=1}^{T}\langle\pi_t - \pi^*, \mathbf{f}(x_t)\rangle\right] + 2|\mathcal{A}|\sqrt{T\log(\max(|\mathcal{F}|,|\mathcal{G}|)/p)} + 4|\mathcal{A}|pT^2 + \alpha\log(|\mathcal{A}|) + \frac{1}{2}$$

$$\leq 2T + 2|\mathcal{A}|\sqrt{T\log(\max(|\mathcal{F}|,|\mathcal{G}|)/p)} + 4|\mathcal{A}|pT^2 + \alpha\log(|\mathcal{A}|) + \frac{1}{2}, \tag{15}$$

where the last inequality holds since $f(x,a) \leq 1$. The above results imply that $\mathbb{E}[q_{T+1}^2] = O(T)$, which is equivalent to $q_{T+1} = O(\sqrt{T})$. Combining these results, we have

$$\mathbb{E}\left[\sum_{t=1}^{T}\langle\pi_t, \mathbf{g}(x)\rangle\right] \leq \mathbb{E}\left[\sum_{t=1}^{T}\|\mathbf{g}(x) - \check{\mathbf{g}}_t(x)\|\right] + q_{T+1} + 2$$

$$\leq \sqrt{T\sum_{t=1}^{T}\|\mathbf{g}(x) - \check{\mathbf{g}}_t(x)\|^2} + q_{T+1} + 2$$

$$\leq \sqrt{T|\mathcal{A}|\sum_{t=1}^{T}\|\varepsilon_t(x,a,p)\|^2} + q_{T+1} + 2$$

$$\leq \sqrt{T|\mathcal{A}|\log(\max(|\mathcal{F}|,|\mathcal{G}|)/p)} + O(\sqrt{T}),$$

which completes the proof that $\mathcal{V}(T) = \tilde{O}(\sqrt{T})$.

## C. Proof of Theorem 6.1

For the regret analysis in contextual bandits with knapsack constraints, we have established the following decomposition into "regret before stopping" and "regret after stopping":

$$\mathcal{R}_b(T) = \tau\nu^* - \sum_{t=1}^{\tau}\langle\pi_t, \mathbf{f}(x_t)\rangle + [T - \tau]\nu^*.$$

We first analyze the regret before stopping, where we can apply Lemma 5.3 and sum over $\tau$ to get:

$$\mathbb{E}\left[\sum_{t=1}^{\tau}\langle\pi_b^* - \pi_t, \mathbf{f}(x_t)\rangle\right] + \mathbb{E}\left[\sum_{t=1}^{\tau}\Delta_t\right] \leq \mathbb{E}\left[2\sum_{t=1}^{\tau}\|\mathbf{f}(x_t) - \hat{\mathbf{f}}_t(x_t)\|\right] + \mathbb{E}\left[\sum_{t=1}^{\tau}\|\check{\mathbf{g}}_{t-1}(x) - \check{\mathbf{g}}_t(x)\|^2\right]$$

$$+ \mathbb{E}\left[\sum_{t=1}^{\tau}Q_t\langle\pi_b^*, \check{\mathbf{g}}_t(x_t) - \mathbf{g}(x_t)\rangle\right] + \mathbb{E}\left[\sum_{t=1}^{\tau}\alpha(D(\pi_b^*||\pi_{t-1}) - D(\pi_b^*||\pi_t))\right]$$

$$+ \sum_{t=1}^{T}\left(\frac{\mathbb{E}[\langle\pi_t, \check{\mathbf{g}}_t(x)\rangle]^2}{2} - \frac{\mathbb{E}[\langle\pi_{t-1}, \check{\mathbf{g}}_{t-1}(x)\rangle]^2}{2}\right),$$

First, we can conduct a telescoping sum to get:

$$\mathbb{E}\left[\sum_{t=1}^{\tau}\alpha(D(\pi^*||\pi_{t-1}) - D(\pi^*||\pi_t))\right] = \mathbb{E}[\alpha D(\pi^*||\pi_1)] \leq \alpha\log(|\mathcal{A}|),$$

$$\sum_{t=1}^{T}\left(\frac{\mathbb{E}[\langle\pi_t, \check{\mathbf{g}}_t(x)\rangle]^2}{2} - \frac{\mathbb{E}[\langle\pi_{t-1}, \check{\mathbf{g}}_{t-1}(x)\rangle]^2}{2}\right) = \frac{\mathbb{E}[\langle\pi_\tau, \check{\mathbf{g}}_\tau(x)\rangle]^2}{2} \leq \frac{1}{2},$$

Then the remaining terms can be similarly bounded by the learning oracle error, which gives

$$\mathbb{E}\left[2\sum_{t=1}^{\tau}\|\mathbf{f}(x_t) - \hat{\mathbf{f}}_t(x_t)\|\right] \leq \mathbb{E}\left[2\sum_{t=1}^{T}\|\mathbf{f}(x_t) - \hat{\mathbf{f}}_t(x_t)\|\right] = \tilde{O}(\sqrt{T})$$

$$\mathbb{E}\left[\sum_{t=1}^{\tau}\|\check{\mathbf{g}}_{t-1}(x) - \check{\mathbf{g}}_t(x)\|^2\right] \leq 2|\mathcal{A}|\mathbb{E}\left[\sum_{t=1}^{T}\|\varepsilon_t(x,a,p)\|^2\right] = \tilde{O}(1)$$

$$
\begin{aligned}
\mathbb{E}\left[\sum_{t=1}^{\tau}Q_t\langle\pi^*,\check{\mathbf{g}}_t(x_t) - \mathbf{g}(x_t)\rangle\right] &= \mathbb{E}\left[\sum_{t=1}^{\tau}Q_t\langle\pi^*,\check{\mathbf{g}}_t(x_t) - \mathbf{g}(x_t)\rangle|\mathcal{E}\right] + \mathbb{E}\left[\sum_{t=1}^{\tau}Q_t\langle\pi^*,\check{\mathbf{g}}_t(x_t) - \mathbf{g}(x_t)\rangle|\bar{\mathcal{E}}\right]\\
&= \mathbb{E}\left[\sum_{t=1}^{\tau}Q_t\langle\pi^*,\check{\mathbf{g}}_t(x_t) - \mathbf{g}(x_t)\rangle|\bar{\mathcal{E}}\right]\\
&\leq \mathbb{E}\left[\sum_{t=1}^{\tau}\|Q_t\|\|\pi^*\|\|\check{\mathbf{g}}_t(x_t) - \mathbf{g}(x_t)\||\bar{\mathcal{E}}\right]\\
&\leq 4|\mathcal{A}|pT^2 = \tilde{O}(1),
\end{aligned}
$$

Combining these terms, we prove an upper bound for regret before stopping:

$$\tau\nu^* - \sum_{t=1}^{\tau}\langle\pi_t,\mathbf{f}(x_t)\rangle = \mathbb{E}\left[\sum_{t=1}^{\tau}\langle\pi_b^* - \pi_t,\mathbf{f}(x_t)\rangle\right] = \tilde{O}(\sqrt{T}).$$

Next, we prove the part of the regret after stopping. From the update rule of $Q_t$, we can immediately obtain the following inequality:

$$q_{\tau+1} + \tau b \geq \mathbb{E}\left[\sum_{t=1}^{\tau}\langle\pi_t,\check{\mathbf{g}}_t(x)\rangle\right],$$

and, using the definition of stopping time, we know that $\tau$ satisfies $q_{\tau+1} + \tau b \geq B := Tb$, which leads to the conclusion that

$$\mathbb{E}[(T-\tau)\nu^*] \leq \frac{\nu^*}{b}\mathbb{E}[q_{\tau+1}].$$

From (15), we can derive that for any $t \in [T]$, we have $\mathbb{E}[q_t] = \tilde{O}(\sqrt{T})$. Setting $t = \tau$, and combining all these terms, we complete the proof.

## D. Proof of Regret and Violation for MAB with stochastic and adversarial constraints

As discussed above, we have `Optimistic`[3] to be readily applied to MAB with stochastic and adversarial constraints. The only difference is that we need to plug in dedicated estimators for this setting. In particular, since our target is to handle the adversarial reward, we use the EXP-3 style estimator for reward

$$\hat{r}_t(a) = \frac{r_t(a)}{\pi_t(a)}\mathbb{I}(a = a_t). \tag{16}$$

For the constraints, we use a weighted empirical estimation in (Bernasconi et al., 2024a) such that it can adapt to both stochastic and adversarial settings. In particular, we design the cost estimator $\bar{c}_t(a)$ and the bonus term $b_t(a)$ such that $\check{c}_t(a)$ satisfies

$$\check{c}_t(a) \leq \bar{c}_t(a) - b_t(a).$$

The bonus term is as the classical upper/lower confidence terms $b_t(a) = \sqrt{\frac{\gamma\log T}{n_t(a)}}$.

Let $\mathcal{T}_{t,a} = \{\tau \leq t, a_t = a\}$ be the set of rounds in which the algorithm plays the action $a$:

$$\bar{c}_t(a) = \sum_{\tau\in\mathcal{T}_{t,a}}w_\tau(a)c_\tau(a) \tag{17}$$

The weights $\{w_\tau(a)\}$ need a careful design such that it is good to track the "slow" change in the stochastic setting and the "fast" change in the adversarial setting. In particular, motivated by online learning, (Bernasconi et al., 2024a) designs the following weights

$$w_\tau(a) = \eta_\tau(a) \prod_{s > \tau, \ s \in \mathcal{T}_{t,a}} (1 - \eta_s(a)) \ \text{ with } \ \eta_\tau(a) = \frac{1 + \Gamma_\tau}{n_{\tau,a}}. \tag{18}$$

where $n_{\tau,a}$ is the number of action $a$ taken by the algorithm and $\Gamma_\tau = [\sum_{s=1}^\tau c_s(a_s) - \nu_{\max}\sqrt{KT}]_0^{\nu_{\max}\sqrt{KT}}$ is the clipped constraint violation to regulate the learning rate. This design is motivated by the online gradient descent to regard $(\bar{c}_t(a) - c_t(a))^2/2$ as the loss function and update as follows

$$\bar{c}_{t+1}(a) = \begin{cases} \bar{c}_t(a) - \eta_t(a)(\bar{c}_t(a) - c_t(a)), & \text{if } a = a_t, \\ \bar{c}_t(a), & \text{if } a \neq a_t. \end{cases}$$

Intuitively, let $\eta_t(a) = \frac{1}{n_t(a)}$ and $\bar{c}_t(a) = \frac{\sum_{\tau \in \mathcal{T}_{t,a}} c_\tau(a)}{n_t(a)}$ become the empirical mean, which works for the stochastic setting. To capture the "fast" change of adversarial setting, the violation $\Gamma_\tau$ is imposed to pay more attention to the recent samples. Then we can construct an LCB-type estimator as

$$\check{c}_t(a) = \bar{c}_t(a) - b_t(a). \tag{19}$$

## `Optimistic`[3] for MAB with Stochastic and Adversarial Constraints

**Initialization:** $\pi_0$, $Q_1 = 0$, $\alpha \geq 2$ and the regularization norm $D(\cdot||\cdot)$.

For $t = 1, \cdots, T - 1$,

- **Optimistic Learning:** in (16) and (19).

- **Optimistic Decision:** Construct policy $\pi_t$ to minimize the surrogate function:

$$\pi_t = \arg\min_\pi -\langle \pi, \hat{\mathbf{r}}_t \rangle + Q_t \langle \pi, \check{\mathbf{c}}_t \rangle + \alpha D(\pi || \pi_{t-1}), \tag{20}$$

  Sample action $a_t \sim \pi_t$.

- **Observe Feedback:** Observe noisy reward $r_t(a_t)$ and cost $c_t(a_t)$.

- **Optimistic Dual Update:** Update the optimistic dual variable $Q_{t+1}$ as follows:

$$Q_{t+1} = [Q_t - \langle \pi_{t-1}, \check{\mathbf{c}}_{t-1} \rangle + 2\langle \pi_t, \check{\mathbf{c}}_t \rangle]^+ \tag{21}$$

**Lemma D.1.** *Let $\pi_t$ be the policy decided by (20), define $q_t = Q_t - \langle \pi_{t-1}, \check{\mathbf{c}}_{t-1} \rangle$ and its corresponding drift $\Delta_t = \frac{1}{2}(q_{t+1}^2 - q_t^2)$. For any feasible policy $\pi$, `Optimistic`[3] achieves*

$$\mathbb{E}[\langle \pi - \pi_{t-1}, \mathbf{r}_{t-1} \rangle] + \mathbb{E}[\Delta_t] \leq \mathbb{E}\left[Q_t \langle \pi, \check{\mathbf{c}}_t \rangle\right] + \alpha\mathbb{E}[D(\pi || \pi_{t-1}) - D(\pi || \pi_t)]$$

$$+ \mathbb{E}[\|\check{\mathbf{c}}_{t-1} - \check{\mathbf{c}}_t\|^2] + \frac{1}{2}\mathbb{E}\left[\langle \pi_t, \check{\mathbf{c}}_t \rangle^2 - \langle \pi_{t-1}, \check{\mathbf{c}}_{t-1} \rangle^2\right] + \frac{1}{\alpha}\mathbb{E}[\|\mathbf{r}_{t-1}\|^2].$$

*Proof.* The proof is almost very similar except we need to pay some attention to the reward part. Note we have $r_{t-1} = \mathbb{E}[\hat{r}_t]$. We have

$$\langle \pi_{t-1} - \pi_t, \hat{\mathbf{r}}_t \rangle + Q_t \langle \pi_t, \check{\mathbf{c}}_t \rangle + \alpha D(\pi_t || \pi_{t-1})$$
$$\leq \langle \pi_{t-1} - \pi, \hat{\mathbf{r}}_t \rangle + Q_t \langle \pi, \check{\mathbf{c}}_t \rangle + \alpha D(\pi || \pi_{t-1}) - \alpha D(\pi || \pi_t)$$

Take the expectation on both sides of the inequality

$$\mathbb{E}[\langle \pi - \pi_{t-1}, \mathbf{r}_{t-1}\rangle] + \mathbb{E}\langle \pi_{t-1} - \pi_t, \mathbf{r}_{t-1}\rangle] + \alpha D(\pi_t||\pi_{t-1}) + \mathbb{E}[Q_t\langle \pi_t, \check{\mathbf{c}}_t\rangle]$$
$$\leq \mathbb{E}[Q_t\langle \pi, \check{\mathbf{c}}_t\rangle] + \alpha\mathbb{E}[D(\pi||\pi_{t-1}) - D(\pi||\pi_t)]$$

Then we leverage the local normal analysis of online mirror descent in (Sun et al., 2017) such that

$$\langle \pi_{t-1} - \pi_t, \mathbf{r}_{t-1}\rangle + \alpha D(\pi_t||\pi_{t-1}) + \frac{1}{\alpha}\|\mathbf{r}_{t-1}\|^2 \geq 0.$$

Therefore, we have

$$\mathbb{E}[\langle \pi - \pi_{t-1}, \mathbf{r}_{t-1}\rangle] + \mathbb{E}[Q_t\langle \pi_t, \check{\mathbf{c}}_t\rangle]$$
$$\leq \mathbb{E}[Q_t\langle \pi, \check{\mathbf{c}}_t\rangle] + \alpha\mathbb{E}[D(\pi||\pi_{t-1}) - D(\pi||\pi_t)] + \frac{1}{\alpha}\mathbb{E}[\|\mathbf{r}_{t-1}\|^2].$$

Following the exact steps to handle the virtual queue and the constraints in Appendix A, we have two key inequalities from Lemmas 5.6 and 5.7

$$\Delta_t \leq q_t\langle \pi_t, \check{\mathbf{c}}_t\rangle + \langle \pi_t, \check{\mathbf{c}}_{t-1}\rangle^2$$
$$= Q_t\langle \pi_t, \check{\mathbf{c}}_t\rangle - \langle \pi_{t-1}, \check{\mathbf{c}}_{t-1}\rangle\langle \pi_t, \check{\mathbf{c}}_t\rangle + \langle \pi_t, \check{\mathbf{c}}_t\rangle^2.$$

and

$$\langle \pi_{t-1}, \check{\mathbf{c}}_{t-1}\rangle\langle \pi_t, \check{\mathbf{c}}_t\rangle \geq \frac{\langle \pi_t, \check{\mathbf{c}}_t\rangle^2}{2} + \frac{\langle \pi_{t-1}, \check{\mathbf{c}}_{t-1}\rangle^2}{2} - \|\check{\mathbf{c}}_{t-1} - \check{\mathbf{c}}_t\|^2 - \|\pi_t - \pi_{t-1}\|^2.$$

Combine these inequalities, we have

$$\mathbb{E}[\langle \pi - \pi_{t-1}, \mathbf{r}_{t-1}\rangle] + \mathbb{E}[\Delta_t] \leq \mathbb{E}[Q_t\langle \pi, \check{\mathbf{c}}_t\rangle] + \alpha\mathbb{E}[D(\pi||\pi_{t-1}) - D(\pi||\pi_t)]$$
$$+ \mathbb{E}[\|\check{\mathbf{c}}_{t-1} - \check{\mathbf{c}}_t\|^2] + \frac{1}{2}\mathbb{E}\left[\langle \pi_t, \check{\mathbf{c}}_t\rangle^2 - \langle \pi_{t-1}, \check{\mathbf{c}}_{t-1}\rangle^2\right] + \frac{1}{\alpha}\mathbb{E}[\|\mathbf{r}_{t-1}\|^2].$$

$\square$

**Lemma D.2.** $\mathtt{Optimistic}^3$ *achieves*

$$\mathbb{E}\left[\sum_{t=1}^T \|\check{\mathbf{c}}_{t-1} - \check{\mathbf{c}}_t\|^2\right] \leq 16 + 16\nu_{\max}\sqrt{KT} + 2\gamma\log^2 T.$$

*Proof.* Let's focus on the difference of $\check{c}_{t-1}(a) - \check{c}_t(a)$ when $a = a_t$ (otherwise the difference is zero)

$$(\check{c}_{t-1}(a) - \check{c}_t(a))^2 \leq 2(\bar{c}_{t-1}(a) - \bar{c}_t(a))^2 + 2(b_{t-1}(a) - b_t(a))^2, \tag{22}$$

where

$$(\bar{c}_{t-1}(a) - \bar{c}_t(a))^2 = \eta_{t-1}^2(a)(\bar{c}_{t-1}(a) - c_{t-1}(a))^2 \leq \frac{4(1 + \Gamma_{t-1})^2}{n_{t-1}^2(a)},$$

$$(b_{t-1}(a) - b_t(a))^2 = \gamma\log T\left(\frac{1}{\sqrt{n_{t-1}(a)}} - \frac{1}{\sqrt{n_t(a)}}\right)^2 \leq \frac{\gamma\log T}{n_{t-1}(a)}.$$

Note $\sum_a n_t(a) = t$. This implies that

$$\mathbb{E}\left[\sum_{t=1}^T \|\check{\mathbf{c}}_{t-1} - \check{\mathbf{c}}_t\|^2\right] \leq \mathbb{E}\left[\sum_{t=1}^T \frac{16 + 16\Gamma_{t-1}^2}{n_{t-1}^2(a_t)} + \sum_{t=1}^T \frac{2\gamma\log T}{n_{t-1}(a_t)}\right]$$
$$\leq \mathbb{E}\left[\sum_{t=1}^T \frac{16 + 16\Gamma_{t-1}^2}{n_{t-1}^2(a_t)}\right] + 2\gamma\log^2 T$$
$$\leq 16 + 16\nu_{\max}\sqrt{KT} + 2\gamma\log^2 T$$

where the first inequality holds because of $(a + b)^2 \leq 2(a^2 + b^2)$; the second inequality holds because the upper bound will be attained when $n_t(a) = \lfloor t/K \rfloor, \forall a \in [K]$ at time $t$; the last inequality holds similarly due to the definition of $\Gamma_\tau = [\sum_{s=1}^\tau c_s(a_s) - \nu_{\max}\sqrt{KT}]_0^{\nu_{\max}\sqrt{KT}}$ and when $n_t(a) = \lfloor t/K \rfloor, \forall a \in [K]$ at time $t$. Specifically, we use

$$
\begin{aligned}
\sum_{t=1}^T \frac{\Gamma_{t-1}^2}{n_{t-1}^2(a_t)} &\leq \sum_{t=1}^T \frac{([t - \nu_{\max}\sqrt{KT}]_0^{\nu_{\max}\sqrt{KT}})^2}{t^2} \\
&= \sum_{t=1}^{\lfloor \nu_{max}\sqrt{KT} \rfloor} \frac{([t - \nu_{\max}\sqrt{KT}]_0^{\nu_{\max}\sqrt{KT}})^2}{t^2} + \sum_{t=\lfloor \nu_{max}\sqrt{KT} \rfloor+1}^T \frac{([t - \nu_{\max}\sqrt{KT}]_0^{\nu_{\max}\sqrt{KT}})^2}{t^2} \\
&= \sum_{t=\lfloor \nu_{max}\sqrt{KT} \rfloor+1}^T \frac{([t - \nu_{\max}\sqrt{KT}]_0^{\nu_{\max}\sqrt{KT}})^2}{t^2} \\
&\leq \sum_{t=\lfloor \nu_{max}\sqrt{KT} \rfloor+1}^T \frac{\nu_{max}^2 KT}{t^2} \\
&\leq \frac{\nu_{max}^2 KT}{\nu_{max}\sqrt{KT}} = \nu_{max}\sqrt{KT}
\end{aligned}
$$

$\square$

Based on Lemmas D.1 and D.2, we have

$$
\mathbb{E}[\langle \pi - \pi_{t-1}, \mathbf{r}_{t-1} \rangle] + \mathbb{E}[\Delta_t] \leq \mathbb{E}[Q_t \langle \pi, \check{\mathbf{c}}_t \rangle] + \alpha \mathbb{E}[D(\pi \| \pi_{t-1}) - D(\pi \| \pi_t)] + \frac{1}{2}\mathbb{E}\left[\langle \pi_t, \check{\mathbf{c}}_t \rangle^2 - \langle \pi_{t-1}, \check{\mathbf{c}}_{t-1} \rangle^2\right]
$$
$$
+ \frac{1}{\alpha}\mathbb{E}[\|\mathbf{r}_{t-1}\|^2] + 16 + 16\nu_{\max}\sqrt{KT} + 2\gamma \log^2 T. \tag{23}
$$

The inequality in (23) is the key and unified result to establish both sublinear regret and violation for MAB with stochastic constraint and $\beta$-regret for MAB with adversarial constraint. As discussed before, we need to impose different baselines for stochastic and adversarial settings.

Recall the offline problem for MAB

$$
\max_\pi \; \mathbb{E}\left[\sum_{t=1}^T \langle \pi, \mathbf{r}_t \rangle\right] \tag{24}
$$

$$
\text{s.t.} \; \mathbb{E}\left[\sum_{t=1}^T \langle \pi, \mathbf{c}_t \rangle\right] \leq 0. \tag{25}
$$

We just need to impose different baselines for these two settings. Intuitively, for the stochastic constraint, we can compare it with the optimal offline policy $\pi^*$ to (24)–(25); for the adversarial constraint, we only compare with a weak $\beta$–optimal offline policy $\beta$–$\pi^*$.

### D.1. `Optimistic`[3] for MAB with Stochastic Constraint

When the constraints are stochastic, i.e., $\mathbb{E}[\mathbf{c}_t] = \mathbf{c}, \forall t \in [T]$, the offline problem is equivalent to

$$
\max_\pi \; \mathbb{E}[\sum_{t=1}^T \langle \pi, \mathbf{r}_t \rangle] \tag{26}
$$

$$
\text{s.t.} \; \mathbb{E}[\langle \pi, \mathbf{c} \rangle] \leq 0. \tag{27}
$$

Based on the key inequality (23), we let $\pi = \pi^*$ be the optimal offline policy to (26)–(27). Therefore, we have the key term to be non-positive

$$
Q_t \mathbb{E}[\langle \pi^*, \check{\mathbf{c}}_t \rangle | \mathcal{H}_t] \leq Q_t \mathbb{E}[\langle \pi^*, \mathbf{c} \rangle | \mathcal{H}_t] = 0.
$$

Therefore, we have

$$\mathbb{E}[\langle \pi^* - \pi_{t-1}, \mathbf{r}_{t-1}\rangle] + \mathbb{E}[\Delta_t] \leq \alpha\mathbb{E}[D(\pi^*||\pi_{t-1}) - D(\pi^*||\pi_t)] + \frac{1}{2}\mathbb{E}\left[\langle \pi_t, \check{\mathbf{c}}_t\rangle^2 - \langle \pi_{t-1}, \check{\mathbf{c}}_{t-1}\rangle^2\right]$$
$$+ \frac{1}{\alpha}\mathbb{E}[\|\mathbf{r}_{t-1}\|^2] + 16 + 16\nu_{\max}\sqrt{KT} + 2\gamma\log^2 T.$$

This implies

$$\mathbb{E}[\sum_{t=1}^{T}\langle \pi^* - \pi_t, \mathbf{r}_t\rangle] + \frac{1}{2}\mathbb{E}[q_{T+1}^2] \leq \alpha\mathbb{E}[D(\pi^*||\pi_0)] + \frac{TK}{\alpha} + \frac{1}{2}\mathbb{E}\left[\langle \pi_T, \check{\mathbf{c}}_{T+1}\rangle^2\right] + 16 + 16\nu_{\max}\sqrt{KT} + 2\gamma\log^2 T$$

$$\leq \alpha\log K + \frac{TK}{\alpha} + 17 + 16\nu_{\max}\sqrt{KT} + 2\gamma\log^2 T.$$

Recall $\alpha = \sqrt{TK/\log K}$, we eventually have

$$\mathbb{E}[\sum_{t=1}^{T}\langle \pi^* - \pi_t, \mathbf{r}_t\rangle] + \frac{1}{2}\mathbb{E}[q_{T+1}^2] \leq 2\sqrt{TK\log K} + 17 + 16\nu_{\max}\sqrt{KT} + 2\gamma\log^2 T. \tag{28}$$

### D.1.1. REGRET ANALYSIS

From (28), we directly have

$$\mathcal{R}(T) \leq \sqrt{TK\log K} + 17 + 16\nu_{\max}\sqrt{KT} + 2\gamma\log^2 T.$$

### D.1.2. VIOLATION ANALYSIS

From (28), we have

$$\mathbb{E}[q_{T+1}^2] \leq 2T + 2(2\sqrt{TK\log K} + 17 + 16\nu_{\max}\sqrt{KT} + 2\gamma\log^2 T).$$

This implies that

$$\mathbb{E}[q_{T+1}] \leq 2\sqrt{T} + 2\sqrt{\sqrt{TK\log K} + 17 + 16\nu_{\max}\sqrt{KT} + 2\gamma\log^2 T}.$$

According to the definition of $q_t = Q_t - \langle \pi_{t-1}, \check{\mathbf{c}}_{t-1}\rangle$ and the update of $Q_t$ in (21), we have

$$\sum_{t=1}^{T}\langle \pi_t, \check{\mathbf{c}}_t\rangle \leq Q_{T+1} + 2 \leq 3 + 2\sqrt{T} + 2\sqrt{\sqrt{TK\log K} + 17 + 16\nu_{\max}\sqrt{KT} + 2\gamma\log^2 T}$$

Now, we proceed to bound the constraint violation, where with high probability,

$$\sum_{t=1}^{T}\langle \pi_t, \mathbf{c}\rangle = \sum_{t=1}^{T}\langle \pi_t, \mathbf{c} - \check{\mathbf{c}}_t\rangle + \sum_{t=1}^{T}\langle \pi_t, \check{\mathbf{c}}_t\rangle$$

$$= \sum_{t=1}^{T}\langle \pi_t, \mathbf{c} - \bar{\mathbf{c}}_t\rangle + \sum_{t=1}^{T}\langle \pi_t, \mathbf{b}_t\rangle + \sum_{t=1}^{T}\langle \pi_t, \check{\mathbf{c}}_t\rangle$$

$$\leq 3\sum_{t=1}^{T}\langle \pi_t, \mathbf{b}_t\rangle + \sum_{t=1}^{T}\langle \pi_t, \check{\mathbf{c}}_t\rangle$$

$$\leq 6\sqrt{T\log T} + 3 + 2\sqrt{T} + 2\sqrt{\sqrt{TK\log K} + 17 + 16\nu_{\max}\sqrt{KT} + 2\gamma\log^2 T},$$

where the first inequality holds from Bernasconi et al. (2024a, Lemma 6.2) that the gap between the empirical mean and (17) is also bounded by $b_t(a)$, and the last inequality holds due to the definition of $b_t(a)$ and the upper bound of estimated violation. Then we proved $\mathcal{V}(T) = \mathbb{E}\left[\sum_{t=1}^{T}\langle \pi_t, \mathbf{c}\rangle\right] = \tilde{O}(\sqrt{T})$.

## D.2. `Optimistic`[3] for MAB with Adversarial Constraint

When the constraints are adversarial, we can only impose a weak baseline that mixes between the unconstrained optimal policy $\pi^*_{adv}$ and a null policy $\pi_{null}$ that minimizes the costs by ignoring the rewards. The trade-off is controlled by a competitive ratio $\beta$, then the mixed policy is $\pi_{mix} = \beta \cdot \pi^*_{adv} + (1-\beta) \cdot \pi_{null}$. As in the setting of stochastic constraints, we study the key term $Q_t \mathbb{E}\left[\langle \pi_{mix}, \check{\mathbf{c}}_t \rangle | \mathcal{H}_t\right]$ and hope to find a mixed policy such that it is non-positive.

According to the definition of $\check{\mathbf{c}}_t = \bar{\mathbf{c}}_t - \mathbf{b}_t$, we have

$$\mathbb{E}\left[\langle \pi_{mix}, \check{\mathbf{c}}_t \rangle | \mathcal{H}_t\right] \leq \mathbb{E}\left[\langle \pi_{mix}, \bar{\mathbf{c}}_t \rangle | \mathcal{H}_t\right].$$

Let's study the key term of $\langle \pi_{mix}, \bar{\mathbf{c}}_t \rangle$ as follows

$$
\begin{aligned}
\langle \pi_{mix}, \bar{\mathbf{c}}_t \rangle &= \sum_a \pi_{mix}(a) \sum_{\tau \in \mathcal{T}_{t,a}} w_\tau(a) c_\tau(a) \\
&= \beta \sum_a \pi^*_{adv}(a) \sum_{\tau \in \mathcal{T}_{t,a}} w_\tau(a) c_\tau(a) + (1-\beta) \sum_a \pi_{null}(a) \sum_{\tau \in \mathcal{T}_{t,a}} w_\tau(a) c_\tau(a) \\
&\leq \beta \sum_a \pi^*_{adv}(a) \sum_{\tau \in \mathcal{T}_{t,a}} w_\tau(a) + (1-\beta)(-\rho) \sum_{\tau \in \mathcal{T}_{t,a}} w_\tau(a) \\
&= \beta + (1-\beta)(-\rho)
\end{aligned}
\tag{29}
$$

where the third inequality holds because $c_\tau(a) \leq 1, \forall a, \tau$ and $c_\tau(a_{null}) = c(a_{null}) = -\rho$. the last equality holds because $\sum_{\tau \in \mathcal{T}_{t,a}} w_\tau(a) = 1$; Finally, recall the definition for competitive ratio $\beta = \rho/(1+\rho)$ such that $\langle \pi_{mix}, \bar{\mathbf{c}}_t \rangle \leq 0$.

Therefore, we have

$$\mathbb{E}\left[\sum_{t=1}^T \langle \pi_{mix} - \pi_t, \mathbf{r}_t \rangle\right] + \frac{1}{2}\mathbb{E}\left[q_{T+1}^2\right] \leq 2\sqrt{TK \log K} + 17 + 16\nu_{\max}\sqrt{KT} + 2\gamma \log^2 T. \tag{30}$$

### D.2.1. REGRET ANALYSIS

Obviously, we have $\mathbb{E}[\sum_{t=1}^T \langle \pi_{mix}, \mathbf{r}_t \rangle] \geq \mathbb{E}[\sum_{t=1}^T \langle \frac{\rho}{1+\rho}\pi^*_{adv}, \mathbf{r}_t \rangle]$, Then we can prove the regret directly from (30),

$$\beta\text{-}\mathcal{R}(T) = \mathbb{E}\left[\sum_{t=1}^T \langle \frac{\rho}{1+\rho}\pi^*_{adv} - \pi_t, \mathbf{r}_t \rangle\right] \leq \sqrt{TK \log K} + 17 + 16\nu_{\max}\sqrt{KT} + 2\gamma \log^2 T.$$

### D.2.2. VIOLATION ANALYSIS

From (30), we have

$$\mathbb{E}[q_{T+1}^2] \leq 2T + 2(2\sqrt{TK \log K} + 17 + 16\nu_{\max}\sqrt{KT} + 2\gamma \log^2 T).$$

This implies that

$$\mathbb{E}[q_{T+1}] \leq 2\sqrt{T} + 2\sqrt{\sqrt{TK \log K} + 17 + 16\nu_{\max}\sqrt{KT} + 2\gamma \log^2 T}.$$

According to the definition of $q_t = Q_t - \langle \pi_{t-1}, \check{\mathbf{c}}_{t-1} \rangle$ and the update of $Q_t$ in (21), we have

$$\mathbb{E}\left[\sum_{t=1}^T \langle \pi_t, \check{\mathbf{c}}_t \rangle\right] \leq \mathbb{E}\left[Q_{T+1}\right] + 2 \leq 3 + 2\sqrt{T} + 2\sqrt{\sqrt{TK \log K} + 17 + 16\nu_{\max}\sqrt{KT} + 2\gamma \log^2 T}$$

Now, we proceed to bound the constraint violation

$$
\begin{aligned}
\mathcal{V}(T) = \mathbb{E}\left[\sum_{t=1}^T \langle \pi_t, \mathbf{c}_t \rangle\right] &= \mathbb{E}\left[\sum_{t=1}^T \langle \pi_t, \mathbf{c}_t - \check{\mathbf{c}}_t \rangle + \sum_{t=1}^T \langle \pi_t, \check{\mathbf{c}}_t \rangle\right] \\
&= \mathbb{E}\left[\sum_{t=1}^T \langle \pi_t, \mathbf{c}_t - \bar{\mathbf{c}}_t \rangle + \sum_{t=1}^T \langle \pi_t, \mathbf{b}_t \rangle + \sum_{t=1}^T \langle \pi_t, \check{\mathbf{c}}_t \rangle\right]
\end{aligned}
$$

Note $b_t(a) = 3\sqrt{\frac{\log T}{n_t(a)}}$, we have

$$\mathbb{E}\left[\sum_{t=1}^T \langle \pi_t, \mathbf{b}_t \rangle \right] \le 3\sqrt{T \log T},$$

where the second inequality holds because the upper bound will be attained when $n_t(a) = \lfloor t/K \rfloor, \forall a \in [K]$ at time $t$. Therefore, we have

$$\mathcal{V}(T) \le \mathbb{E}\left[\sum_{t=1}^T \langle \pi_t, \mathbf{c}_t - \bar{\mathbf{c}}_t \rangle \right] + 3 + 5\sqrt{T \log T} + 2\sqrt{\sqrt{TK \log K} + 17 + 16\nu_{\max}\sqrt{KT}} + 2\gamma \log^2 T$$

Let's focus on the key term $\sum_{t=1}^T \langle \pi_t, \mathbf{c}_t - \check{\mathbf{c}}_t \rangle$. Here recall an important corollary in Bernasconi et al. (2024a, Corollary 5.7), from the fact that the learning $\eta_t(a)$ rates are non-increasing, which gives

$$\mathbb{E}\left[\sum_{t=1}^T \langle \pi_t, \mathbf{c}_t - \bar{\mathbf{c}}_t \rangle \right] = \mathbb{E}\left[\sum_a \sum_{t \in \mathcal{T}_{T,a}} \frac{\bar{c}_{t+1}(a) - \bar{c}_t(a)}{\eta_t(a)} \right] \le \sum_a \frac{1}{\eta_T(a)},$$

where $\eta_t(a) = (1 + \Gamma_t)/n_{t,a}$ and $\Gamma_t = [\sum_{s=1}^t c_s(a_s) - \nu_{\max}\sqrt{KT}]_0^{\nu_{\max}\sqrt{KT}}$. We provide a simplified proof in the following. Fix action $a$ and let $k = |\mathcal{T}_{T,a}|$ be the number of times action is played before time horizon $T$. Define $t(j)$ be the rounds in which action $a$ is played the $j$-th time. We can obtain:

$$\begin{aligned}
&\sum_{t \in \mathcal{T}_{T,a}} \frac{\bar{c}_{t+1}(a) - \bar{c}_t(a)}{\eta_t(a)} \\
&= \sum_{j \in [k-1]} \frac{1}{\eta_{t(j)}(a)}(\bar{c}_{t(j+1)}(a) - \bar{c}_{t(j)}(a)) + \frac{1}{\eta_{t(k)}(a)}(\bar{c}_{t(k)+1}(a) - \bar{c}_{t(k)}(a)) \\
&\le \sum_{j \in [k-1]} \left( \frac{1}{\eta_{t(j+1)}(a)}\bar{c}_{t(j+1)}(a) - \frac{1}{\eta_{t(j)}(a)}\bar{c}_{t(j)}(a) \right) + \frac{1}{\eta_{t(k)}(a)}(\bar{c}_{t(k)+1}(a) - \bar{c}_{t(k)}(a)) \quad (31) \\
&= \frac{1}{\eta_{\tau(k)}(a)}\bar{c}_{t(k)+1}(a) - \frac{1}{\eta_{\tau(1)}(a)}\bar{c}_{t(1)}(a) \\
&\le \frac{1}{\eta_{\tau(k)}(a)}\bar{c}_{t(k)+1}(a) \le \frac{1}{\eta_T(a)}.
\end{aligned}$$

The above proof requires that the costs (or the estimated ones) are non-negative in the adversarial setting. Specifically, the key inequality (31) requires not only that the learning rate sequence $\eta_t(a)$ be non-increasing, but also that $\bar{c}_t(a) \ge 0$ for all $t$.

Combining these facts, we have

$$\mathcal{V}(T) \le \frac{KT}{1 + \Gamma_T} + 3 + 5\sqrt{T \log T} + 2\sqrt{\sqrt{TK \log K} + 17 + 16\nu_{\max}\sqrt{KT}} + 2\gamma \log^2 T$$

Now, if $\mathcal{V}(T) \ge 2\nu_{\max}\sqrt{KT \log T}$, we have $\Gamma_T = \nu_{\max}\sqrt{KT \log T}$ such that

$$\begin{aligned}
\mathcal{V}(T) &\le \frac{KT}{2\nu_{\max}\sqrt{KT \log T}} + 3 + 5\sqrt{T \log T} + 2\sqrt{\sqrt{TK \log K} + 17 + 16\nu_{\max}\sqrt{KT}} + 2\gamma \log^2 T \\
&\le 3\nu_{\max}\sqrt{KT \log T}.
\end{aligned}$$

Therefore, it concludes the proof of violation of adversarial constraints.

## E. Additional experiment on BwK with stochastic and adversarial constraints

We evaluate our approach through numerical experiments on non-contextual multi-armed bandits across both stochastic and adversarial environments. We consider $|\mathcal{A}| = 10$ arms and the time horizon $T = 10000$. All results are obtained by averaging over 50 trials and reported with a 95% confidence interval.

- In the stochastic setting, each arm's mean reward $r(a) = \mu_a$ is drawn from the uniform distribution of $[0, 1]$ while mean constraint $c(a) = \lambda_a$ is drawn from the uniform distribution of $[-0.5, 1]$. Observations of rewards and costs are perturbed by Gaussian noise $\mathcal{N}(0, 0.05)$.

- In the adversarial setting, rewards and constraints adopt time-varying dynamics: $r_t(a) = \mu_a + \alpha_a^1 \sin(\omega_a^1 t)$ and $c_t(a) = \lambda_a + \alpha_a^2 \sin(\omega_a^2 t)$, where $\mu_a, \lambda_a$ match the stochastic setting, amplitude and frequencies $\alpha_a^1, \alpha_a^2, \omega_a^1, \omega_a^2 \sim$ Uniform$[0, 0.2]$. Observations of rewards and costs are perturbed by Gaussian noise $\mathcal{N}(0, 0.05)$.

Figures 2 demonstrates that $\texttt{Optimistic}^3$ outperforms all baseline algorithm, which justify our theoretical guarantees.

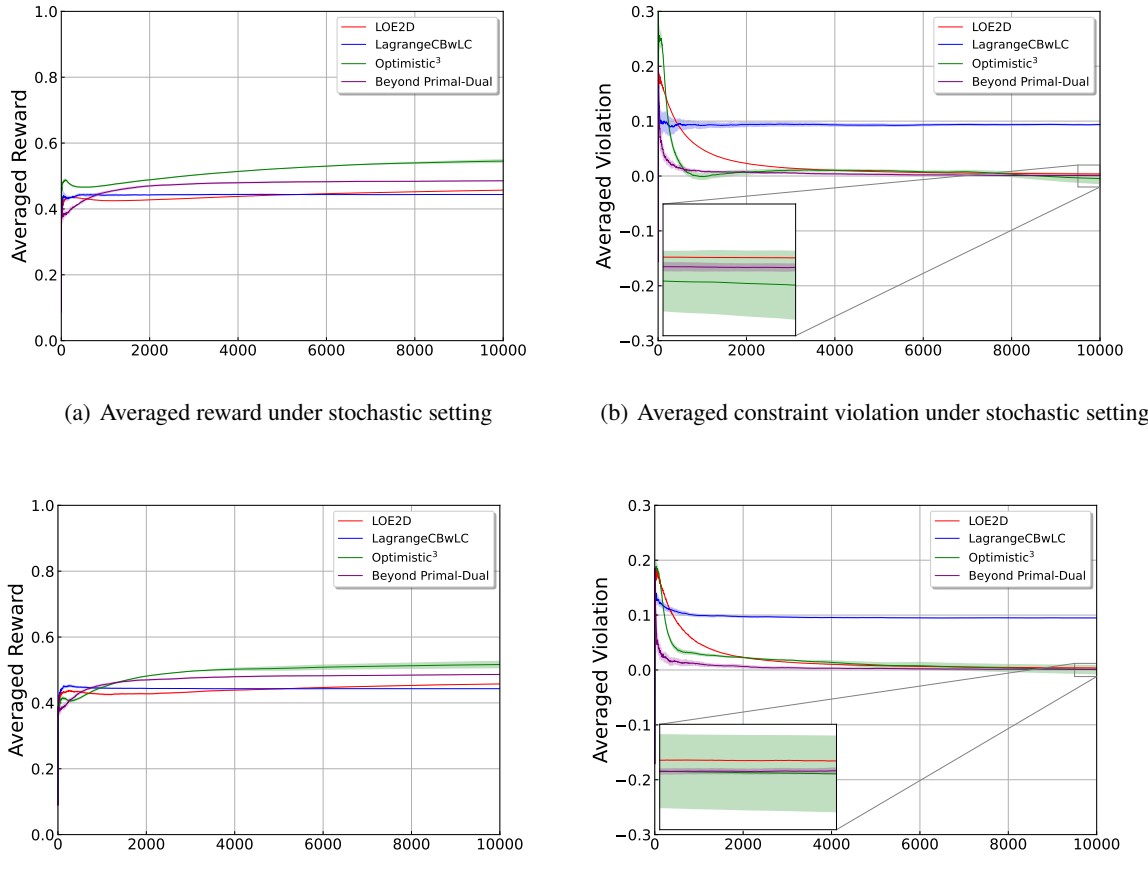

(a) Averaged reward under stochastic setting

(b) Averaged constraint violation under stochastic setting

(c) Averaged reward under adversarial setting

(d) Averaged constraint violation under adversarial setting

*Figure 2.* Averaged reward and constraint violation under LOE2D (Guo & Liu, 2024), LagrangeCBwLC (Slivkins et al., 2023), Beyond Primal-Dual (Bernasconi et al., 2024a) and $\texttt{Optimistic}^3$.

