# OpenReview forum: "Triple-Optimistic Learning for Stochastic Contextual Bandits with General Constraints"
_ICML.cc/2025/Conference — ICML 2025 poster_

### Official Review · Reviewer_38bH · 2025-03-07

**Overall Recommendation:** 2

**Summary:**

The paper introduces Optimistic³, a triple-optimistic framework for stochastic contextual bandits with general constraints. The main contribution is that it achieves regret and constraint violation bounds of without requiring Slater’s condition. It is applicable not only to general constrained contextual bandits but also to classical multi-armed bandit scenarios. Experiments are conducted using the real dataset comparing Optimistic³ with previous approaches, showing empirical advantages.

**Claims And Evidence:**

Yes

**Essential References Not Discussed:**

References are well discussed.

**Experimental Designs Or Analyses:**

Yes, no obvious issue on experiments as the paper is largely theoretical.

**Methods And Evaluation Criteria:**

Yes

**Other Comments Or Suggestions:**

- A minor formatting issue is that the paper largely overlooks in-context citations.
- The table referencing (Guo & Liu, 2024) is confusing due to unclear splitting across rows. I am not sure what they are referring to.
- Including formal mathematical definitions of Slater’s condition and the Slater parameter would enhance clarity.
- Explicitly clarifying the optimality hidden within the \(\tilde{O}(\cdot)\) notation is recommended, as precise descriptions are typically preferred in bandit literature. It is not clearly what is hidden in \(\tilde{O}(\cdot)\) now or if same things are hidden throughout. This is a major clarity issue of this paper, optimal vs nearly optimal.

**Other Strengths And Weaknesses:**

Strengths:
- The theoretical results are robust and generalize well, effectively removing problematic dependencies.
- The references listed are recent and highly relevant.

Weaknesses:
- Please see the suggestions below.

**Questions For Authors:**

Could the authors please provide additional details on the experimental setup? For instance, it would be helpful to explain explicitly how the context is generated. Additionally, is it possible to run a non-contextual version of the experiment to validate the claims made in Section 7?

**Relation To Broader Scientific Literature:**

Its key contributions closely relate to and extend existing literature on both of contextual and non-contextual bandits under constraints.

**Theoretical Claims:**

No due to time limit.

---

> ### Author Rebuttal · Authors · 2025-04-01
>
> We appreciate the reviewer's comments and want to address your major concerns below.
>
> - **Table reference (Guo \& Liu, 2024).** We adopt a row-wise split in referencing their work to properly represent their work, as their unified algorithm delivers distinct theoretical guarantees across two regimes: (i) a Slater-free setting yielding $\tilde{O}(T^{\frac{3}{4}})$ bounds for both regret and constraint violation, and (ii) a Slater-satisfied regime achieving sharper $\tilde{O}(\sqrt{T}/\delta^2)$ guarantees.
>
> - **The Slater's condition.** The Slater condition assumes the existence of a strict feasible solution and is a standard assumption in constrained optimization. While our analysis does not explicitly rely on this condition (its elimination constitutes a key theoretical contribution), we acknowledge the need to formally define it and have included this in our revision.
>
> - **Clarification on $\tilde{O}$.** The notation $\tilde{O}$ suppresses logarithmic factors, which is consistent with the state-of-the-art works. We will explicitly define it in our revision.
>
> - **Experimental setup.** Our experiments replicate the methodology in (Guo \& Liu (2024)), where contexts are randomly sampled from the MSLR dataset to simulate customer arrivals. We have included a more detailed description in our revision.
>   We have conducted additional experiments to verify our results in non-contextual bandits with stochastic and adversarial constraints. Optimistic$^3$ outperforms all baseline algorithms in both settings, which aligns with our theoretical guarantees. Complete implementation details and supporting figures are available at [https://anonymous.4open.science/r/Test-25FA/additional_exp.pdf](https://anonymous.4open.science/r/Test-25FA/additional_exp.pdf).
>
> We hope our responses have clarified the reviewer's concerns and that the reviewer can re-evaluate our work. Please feel free to share any additional feedback, and we will address it thoroughly.

---

### Official Review · Reviewer_cndg · 2025-03-12

**Overall Recommendation:** 3

**Summary:**

The paper proposes a Triple-Optimistic Learning method for contextual bandit with general unknown constraints. And further the paper shows a regret and violation of $\tilde{O}(\sqrt(T))$, which outperforms the previous method on $\tilde{O}(T^{3/4})$.  They also provide the regret bound of CBwK and extend to the adversarial and stochastic bandit.

**Claims And Evidence:**

Some of the writing/assumptions might not be clear and need further clarification.

1, In assumption 3.2, why the sum of UCB/LCB uncertainty $\epsilon$ is bounded by the term in Line 191? Is it from the general UCB algorithm? And further, does UCB and LCB for reward and cost share the same \epsilon? If yes, I believe this could be a wrong assumption?

2, Why in Optimistic Learning part, you are learning the LCB for the cost? I know some of the paper uses UCB for the cost and some uses LCB. Can you provide more details why using LCB? How does it compared to UCB?

3, If I understand correctly, is the $Q_t$ similar to the idea of lagrange multipliers? Does the expectation of $E_x$ taking the expectation over all the historical data?

4, Why uses (7) to find the optimal policy instead of selecting the argmax of the L?

5, Why including the $\Delta_t$ in the regret bound in Lemma 5.3? Where does it come from?

**Essential References Not Discussed:**

The paper doesn't discuss the contextual bandit under stage-wise constraint.

**Experimental Designs Or Analyses:**

Yes. The experiment section makes sense to me.

**Methods And Evaluation Criteria:**

Yes. But only includes one dataset.

**Other Comments Or Suggestions:**

I feel the writing can be more clear, including what is the Slater’s condition, how does it connect to your method. Also, page 6 left column is a bit hard to follow, maybe provide more takeaway or remark will be helpful.

**Other Strengths And Weaknesses:**

See above.

**Questions For Authors:**

See above.

**Relation To Broader Scientific Literature:**

It contributes to the bandit under the safety constraint, provide an algorithm that has a lower regret and violations.

**Theoretical Claims:**

Some of them are mentioned in Claims And Evidence*.

1) why the error of the reward is $\tilde{O}(1)$ in Line 312 Left column.

---

> ### Author Rebuttal · Authors · 2025-04-01
>
> We very appreciate the reviewer for the constructive comments and want to address your major concerns below.
>
> - **Online regression oracle.** The online regression oracle assumption is standard in contextual bandits (Foster et al., 2018; Slivkins et al., 2023; Han et al., 2023; Guo \& Liu, 2024; Chzhen et al., 2024; Anonymous, 2025). This assumption extends the UCB/LCB (upper/lower confidence bound) paradigm to handle diverse environmental settings. The LinearUCB algorithm serves as a fundamental example that inherently satisfies this assumption. For broader applications, [R1] presents an efficient binary search method for UCB/LCB construction.
>
> - **The usage of UCB/LCB.** Both UCB and LCB enhance algorithmic exploration in online learning but serve distinct roles: UCB optimistically guides exploration toward actions with lower estimated rewards but higher uncertainty, prioritizing under-explored regions of the reward space. LCB encourages the exploration of actions that appear marginally unsafe under current estimates (i.e., with higher predicted costs) but may still satisfy constraints upon further investigation. By assigning LCB estimators to cost function, the algorithm temporarily treats these actions as ``safe," enabling exploration of uncertain constraint regions.
>
> - **$Q_t$ and $\mathbb{E}_{x}$.** While $Q_t$ can be interpreted analogously to Lagrange multipliers (or dual variables), its update rule differs fundamentally due to its optimistic design and unbounded property. This design accelerates convergence and enables our key analytical framework to operate without reliance on the Slater condition.
>   The notation $\mathbb{E}_{x}$ denotes expectation only over the context variable (drawn from a known distribution $\mathbb{P}$), not over the entire history.
>   We apologize for any confusion caused by this notation and have clarified it in the revised manuscript.
>
> - **Optimistic decision process.** The surrogate function in (6) represents the estimated scores over actions. While prior works (e.g., Chzhen et al., 2024; Anonymous, 2025) directly optimize these scores, we introduce an additional regularization term in (7) to stabilize and smooth the update since we introduce the optimistic design to speed up dual updates and encourage exploration.
>
> - **The drift term $\Delta_t$.** The Lyapunov drift term $\Delta_t$ quantifies the stability of the virtual queue, enabling us to bound cumulative constraint violations through its analysis (see lines 375–381). This term comes from the direct usage of Lemma 5.6 to replace $Q_t \mathbb{E}_x[\langle\pi_t,\check g_t(x)\rangle]$ in line 293.
>
> - **Other comments.** (i) The cumulative error of reward in line 312 is a typo, and it should be $\tilde{O}(\sqrt{T})$, the proof can be found in line 699. (ii) We will briefly introduce contextual bandits under stage-wise constraints (also termed safe bandits), where algorithms typically assume access to an initial safe region and rely on the iterative construction of safe decision sets. (iii) We sincerely appreciate your suggestions and will incorporate more intuitive explanations and detailed discussions to ensure the methodology and proofs are easy for readers to follow.
>
> We hope that our response addresses the reviewer's concerns and that the reviewer can re-evaluate our work. Please let us know if you have any further comments, and we will try our best to address them.
>
> ---------------------------------------------Reference-------------------------------------------------
>
> [R1] Foster, Dylan, et al. Practical contextual bandits with regression oracles. International Conference on Machine Learning (ICML), 2018.

---

> > ### Comment · Reviewer_cndg · 2025-04-05
> >
> > Thanks for addressing my concerns. I will raise my score to 3.

---

> > > ### Author Response · Authors · 2025-04-05
> > >
> > > We sincerely appreciate your constructive feedback and the time invested in reviewing our work！

---

### Official Review · Reviewer_MfNn · 2025-03-14

**Overall Recommendation:** 4

**Summary:**

$\newcommand{\vg}{\textrm{\v{$g$}}}$
$\newcommand{\ip}[1]{\langle #1\rangle}$

This paper is concerned with contextual finite-action bandits with unknown aggregate constraints. Concretely, in every round, the learner first observes a context $x_t$ drawn iid from a law which is known to the learner, and selects an action $a_t$. As feedback, the learner observes a reward $r_t(x_t,a_t)$ and a cost $c_t(x_t, a_t),$ which are each realisable in that there exist classes $\mathcal{F}, \mathcal{G}$ with good regression oracles such that $\forall t, a, x, \mathbb{E}[ r_t(x_t,a) | x_t = x] = f(x,a), \mathbb{E}[ c_t(x_t,a) | x_t = x] = g(x,a)$ for some $f \in \mathcal{F}$ and $g \in \mathcal{G}$. Note that since the number of arms is finite, these can instead just be written as vectors $f(x)$ and $g(x)$. The goal is to maximise the accumulated reward, while ensuring that the violation $\sum c_t(x_t, a_t)$ is small.

Notice that this setting admits a static offline optimal policy, $\pi^*$ which solves $\max_{\pi} \mathbb{E}[ \langle \pi, f(x)\rangle] \textrm{ s.t. } \mathbb{E}[ \langle \pi, g(x)\rangle \le 0,$ where the expectation is over $x$. The paper proposes a new `triply optimistic' algorithm that ensures that the pseudo-regret $\mathcal{R}\_T :=  T \mathbb{E}[ \langle \pi^*, f(x)\rangle] - \mathbb{E}[ \sum r\_t(x\_t,a\_t)]$ and violations $\mathcal{V}\_T := \mathbb{E}[\sum_t c\_t(x\_t, a\_t)]$ are both bounded as $\tilde{O}(\sqrt{T})$ without knowledge of or dependence on a Slater parameter (i.e., the gap in Slater's condition) for the static problem. This offers a significant improvement in that prior work on this problem either assumes knowledge of the Slater parameter (and incurs an inverse dependence it), or suffers a weakened $T^{3/4}$ regret in the absence of the same.

The algorithm underlying this method first computes optimistic estimates of $f, g,$ i.e., a $\hat{f}_t$ such that $\hat{f}_t \ge f$ and $\vg_t$ such that $\vg_t \le g$ everywhere (which high probability). The policy $\pi_t$ is selected by optimising $\langle \pi, \hat{f}\_t\rangle - Q\_t \langle \pi,\vg\_t\rangle - \alpha D( \pi \| \pi\_{t-1}),$ the final term serving as a regulariser. The most interesting aspect of this method is the choice of the multiplier $Q_t$, which is updated as $ Q\_{t+1} = \max(0, Q\_t + 2 \mathbb{E}\_x[ \langle \pi\_t, \vg\_t(x)\rangle] - \mathbb{E}\_x[ \langle \pi\_{t-1}, \vg\_{t-1}]),$ which the authors term to be an "optimistic adaptation" of the multiplier, viewing $\mathbb{E}\_x [ \langle \pi\_t , \vg\_t\rangle - \langle \pi\_{t-1}, \vg\_{t-1}\rangle]$ as a momentum term. This method is analysed by developing a direct bound on $\mathbb{E}[ \langle \pi^* - \pi\_t, f(x\_t)\rangle] + \mathbb{E}[ q\_{t+1}^2 - q\_t^2],$ where $q\_t = Q\_t - \mathbb{E}\_x[ \langle \pi\_{t}, \vg\_t\rangle],$ which is bounded by a bunch of telescoping terms along with estimation error terms of the form $\\| f(x) - \hat{f}_t(x)\\|$ that are controlled by assumption.

Along with the above result, this same method also yields a bound for the "BwK" case by a standard calculation. The authors also argue (albeit in a fairly terse section) that the same approach can be combined with a (context-free) estimation technique due to a recent work of Bernasconi et al. to yield "best of both worlds" guarantees in bandits with aggregate stochastic _and_ adversarial constraints.

**Claims And Evidence:**

I think the main claim of Theorem 5.1 is incorrect due to a bug in Lemma 5.6. Please see the 'Theoretical Claims' section below.

**Essential References Not Discussed:**

This is fine. Not essential, but I would prefer if the distinction from the "safe bandit" literature (where constraints are enforced in every round) were discussed briefly.

**Experimental Designs Or Analyses:**

---

**Methods And Evaluation Criteria:**

---

**Other Comments Or Suggestions:**

- I understand that you want the name "triple optimism", but I don't see why the setting of $Q_t$ can be interpreted as "optimistic". This choice of name needs to be justified.

- I think section 7 is way too terse of a presentation of these results. Like, I don't think the adversarial problem is even properly set up in the paper, so it's very hard to be certain of the meaning of anything said. Appendix D is similarly terse - I happen to be familiar with the Bernasconi et al. paper, so could plod through, but surely this should be self-contained enough that a reader has a hope of understanding what is being said (especially to do with the estimation, which is the main change). Expanding this should be important.

- I don't really understand why the min-selection fairness constraints are discussed - aren't the $\lambda_i$ known a priori in this problem? I also think the explicit discussion of the group-fairness constraints is somewhat unnecessary unless the paper explicitly addresses the further structure in this problem (in that it is obvious that such a constraint can be written in the form stated).

- A bunch of proofs are somewhat carelessly written, and should be addressed:

    - Line 645: why is $\\|\vg\_t(x)\\| \le 1? $ Aren't you only told that $c\_t \in [-1,1]$. This means that the only reasonable assumption is that $\\|\vg_t\\| \le \sqrt{ |\mathcal{A}|}$, not that it is $\le 1,$ right? Which would in turn bump up $\alpha$? This all still works - use Hoelder's to bound as $\\|\hat{g}\_{t-1}\\|\_{\infty} \cdot \\| \pi_t - \pi\_{t-1}\\|\_{1},$ and hit the second with Pinsker's, but as written, I think it's natural for me to assume that the norm just means 2-norm, and then it's full-on confusion city (and worrying about contagion). Please rewrite this stuff correctly.
    - Line 627: the equality here is only true if $h(\pi_t) = -q_t - \mathbb{E}\_{x} [ \langle \pi_t, \hat{g}_t(x)\rangle].$ Of course, in the other case the inequality is trivial, but this needs to be written, otherwise the proof as stated is incomplete.
    - Line 568: there's a typo: should be $D(\pi\\|\pi\_{\mathrm{opt}})$ on the RHS. Also, while I'm at it, not sure that the $\hat{h}$ notation adds any clarity, since it is always expanded immediately, and in line 572, it's probably worth explicitly invoking convexity of $h$ to write $h(\pi) - h(\pi\_{\mathrm{opt}}) \ge \langle \nabla h(\pi\_{\mathrm{opt}}), \pi - \pi\_{\mathrm{opt}}\rangle,$ which would add some clarity.
    - Line 313 Col 2: the $\sum \\|f - \hat{f}\_t\\|$ should be $O(\sqrt{T})$ instead of $O(1)$?
    - Line 618: I got confused because the main text had already defined a $h(\pi)$. Maybe use a different letter?
    - Line 654: should be $2a^2 + 2b^2$.

**Other Strengths And Weaknesses:**

I think the approach of the paper is interesting, and the results are remarkable, given the nearly decade-long history of the Slater parameter and cost sign assumptions bumping along in the aggregate constraint bandit literature. The augmentation is simple, but quite elegant.

The natural sticking point is that the method requires knowing the context distribution a priori. This is certainly restrictive, but in my opinion the results are valuable nevertheless, particularly when compared to the recent work establishing similar guarantees in the noncontextual setting due to the much simpler algorithmic structure.

I think that with the promise that the paper will be edited both for typos, and to expand the discussion of section 7, I am quite happy to recommend this paper.

**Questions For Authors:**

- I liked the use of the OMD-style regularisation to eliminate the $\\|\pi\_t - \pi_{t-1}\\|_1$ drift term showing up in the analysis. Presumably this can be extended to other Bregman divergences? A short account would be nice (perhaps in an appendix).

**Relation To Broader Scientific Literature:**

This is a nice paper. The approach taken is a simple change to the multiplier design, but has a far-ranging effect by attaining the tight square root regret without dependence on the Slater parameter. This would definitely be of interest to the online learning community at ICML.

**Theoretical Claims:**

The theoretical issues I raised have been clarified by the rebuttal, and to my understanding, the results are correct.

~~I think there is a serious bug in the paper, which breaks the results.~~

~~Firstly, let me note that the notation $\mathbb{E}_x$ was never defined, which is important since expressions like $\mathbb{E}_x[ \ip{ \pi\_t, \vg\_t(x)}]$ are constantly used within the paper. My interpretation of these is that this is the expectation of $\langle \pi_t, \vg_t(Z)\rangle$ where $Z$ is drawn from the same law as the context $x_t$. This is more or less the only reasonable thing I could infer, because this notation appears with both $\vg_t(x)$ and $\vg\_{t-1}(x),$ and of course since it is explicitly said that the knowledge of the law of the context is needed (and I don't see where else this is employed). I think the mistake however would be present even if this expectation were with respect to something else.~~

~~In any case, I think there is a mistake in Lemma 5.6: From line 631, we should get the term $-\mathbb{E}_x[ \langle \pi_t, \vg_t(x)\rangle ]\cdot \mathbb{E}_x[ \langle \pi\_{t-1}, \vg\_{t-1}(x)\rangle ] \neq -\mathbb{E}_x[ \langle \pi_t, \vg_t(x)\rangle \langle \pi\_{t-1}, \vg\_{t-1}(x)\rangle],$ i.e., the product of expectations rather than the expectation of the product. However the statement of Lemma 5.6 has the expectation of products instead.~~

~~This is far from inconsequential: through Lemma 5.7, this expectation of products is utilised to gain terms of the form $- ( \mathbb{E}_x[\ip{\pi_t, \vg_t(x)}^2] + \mathbb{E}\_x[ \ip{\pi\_{t-1}, \vg\_{t-1}(x)}^2 ])/2, $ which interacts with a $+ \mathbb{E}_x[ \ip{\pi_t, \vg_t(x)}^2]$ term to gain a telescoping expression $( \mathbb{E}_x[\ip{\pi_t, \vg_t(x)}^2] - \mathbb{E}\_x[ \ip{\pi\_{t-1}, \vg\_{t-1}(x)}^2 ])/2$ in Lemma 5.3. However, the corrected version (controlling the product of the expectations) would give lower bounds in terms of $\mathbb{E}_x[ \langle \pi_t, \vg_t(x)\rangle]^2$ (square outside the expectation), which means that now instead of this telescoping sum, there's a $ \sum \mathbb{E}_x[ \langle \pi_t, \vg_t(x)\rangle^2] - \mathbb{E}_x[\langle \pi_t, \vg_t(x)\rangle ]^2/2 - \mathbb{E}_x[\langle \pi\_{t-1}, \vg\_{t-1}(x)\rangle]^2/2$ left dangling. Adding and subtracting the appropriate squares of the expectation, this gives a (non-telescoping) overhead of $\frac12\sum \mathrm{Var}_x( \ip{\pi_t, \vg\_t(x)}) + \mathrm{Var}_x(\ip{\pi\_{t-1}, \vg\_{t-1}(x)}) = \sum \mathrm{Var}_x( \ip{\pi_t, \vg_t(x)}) + O(1)$. I don't think this term can be considered sublinear in $T$ (let alone $O(\sqrt{T})$), which means that the bound instead obtained is that $\mathcal{R}_T + q_T^2 \le O(T)$, which gives no control on the regret.~~

----

~~By the way, I think the above might be fixable, in that I don't really see why you need the $\mathbb{E}\_x$ everywhere: suppose that this is dropped, and you operate with $\tilde{Q}\_{t+1} = \max(0, \tilde{Q}\_t + 2\ip{\pi\_t, \vg\_t(x_t)} - \ip{\pi\_{t-1}, \vg\_{t-1}(x_{t-1})})$. Then an appropriate version of Lemma 5.6 does go through (seeing as there is now no expectation operator that is falsely distributed), and thus Lemma 5.7 goes through. The resulting bound is then on $\mathcal{R}\_T + \mathbb{E}[\tilde{q}\_{T+1}^2],$ where $\tilde{q}\_{T+1} = \tilde{Q}\_{T+1} - \ip{\pi\_T, \vg\_T(x_T)}.$ The regret bound is immediate, and for the violation bound, you instead get via a telescope that $\sum \ip{\pi_t, \vg_t(x_t)} \le \tilde{q}\_{T+1} + O(1)$ surely, at which point you can pass to $\sum \ip{\pi_t, g(x_t)}$ and hit things with an expectation, using the bound on $\mathbb{E}[\tilde{q}\_{T+1}^2] \ge \mathbb{E}[ \tilde{q}\_{T+1}]^2$. Of course I might have something wrong here, and even if this works, then the edits should be made and the paper re-reviewed.~~

---

Besides this the analysis appears to be correct, although there are a bunch of carelessly written proofs (see the comments section below).

---

> ### Author Rebuttal · Authors · 2025-04-01
>
> We sincerely appreciate the reviewer’s meticulous review and constructive feedback. The reviewer's concern is correct, where there are big typos in Lemma $5.6$ (and propagated into Lemma $5.7$). We sincerely apologize for the oversight and any confusion caused. Our proof remains valid when these typos are fixed. We are very thankful that the reviewer liked our "optimistic" idea and theoretical results (yes, they are correct). We are also very excited about these results when we realize that "optimistic" design from adversarial online learning can address the long-standing challenges in stochastic bandits. We sincerely invite the reviewer to re-evaluate the paper based on our response.
>
> - **Definition of $\mathbb{E}_x$.** The expectation $\mathbb{E}_x$ is taken w.r.t. the context variable, as correctly noted by the reviewer. Here, the variable $x$ follows the distribution $\mathbb{P}$, identical to the distribution of $x_t$. We will specify the definition in our revision.
>
> - **Typos in Lemma $5.6$ and Lemma $5.7$.**
>   As the reviewer correctly identified, the typo in Lemma 5.6 involves the cross term:
>   $$\mathbb{E}_x[\langle \pi_t,\check g_t(x)\rangle]\mathbb{E}_x[\langle \pi_t,\check g_{t-1}(x)\rangle]$$
>   which is misplaced by $\mathbb{E}_x[\langle \pi_t,\check g_t(x)\rangle\langle \pi_t,\check g_{t-1}(x)\rangle]$. This typo is unfortunately propagated into Lemma 5.7. Therefore, Lemma 5.7 needs to take care of the corrected cross term, and it is indeed correct by fixing the corresponding terms as follows
> \begin{aligned}
> &\mathbb{E}_x [\langle \pi_{t-1}, \check{\mathbf{g}}_{t-1}(x)\rangle] \mathbb{E}_x [\langle \pi_{t}, \check{\mathbf{g}}_{t}(x)\rangle] \\
> \geq  &\frac{\mathbb{E}_x [\langle \pi_{t}, \check{\mathbf{g}}_{t}(x)\rangle]^2}{2} + \frac{\mathbb{E}_x [\langle \pi_{t-1}, \check{\mathbf{g}}_{t-1}(x)\rangle]^2}{2} \\
> & - \mathbb{E}_x [\| \check{\mathbf{g}}_{t-1}(x) - \check{\mathbf{g}}_{t}(x) \|^2] - \|\pi_t - \pi_{t-1}\|_1^2.
> \end{aligned}
>   where we fixed the typo of $\mathbb{E}_x [\langle \pi_{t}, \check{\mathbf{g}}_{t}(x)\rangle^2]$ in the original version.
>   We have attached a complete corrected proof in [https://anonymous.4open.science/r/Pr25/](https://anonymous.4open.science/r/Pr25/). We apologize for the typo again and thank you very much for your time to read the correction.
>
> - **The origin of the typo.**
>   We want to explain that the reason behind the typo is that when we started working on this problem, we indeed tried the ambitious sample-based design, as you suggested. However, this would have issues when establishing Lemma 5.7.
>   Specifically, this design would introduce $$\|\check{g}_t(x_t) - \check{g}_{t-1}(x_{t-1})\|^2,$$ which is extremely challenging to quantify as the difference between two realizations are involved. Then, when we switched to the current "expected" design, we apologize that we had the copy-paste typos.
>
> - **Extension to other Bregman divergences.** This can be extended to other Bregman divergences, and we will include a discussion in the appendix.
>
> - **"Triple-optimism":**
>   The term $Q_t$ is characterized as "optimistic" due to its conceptual roots in the Optimistic Online Mirror Descent framework in [R1], which leverages predictions to accelerate convergence. Inspired by its application to min-max problems in [R2], we designed an optimistic dual update mechanism, accelerates adaptation through a correction term that incorporates predictive insights, unlike traditional methods that rely solely on current feedback and employ a "pessimistic" exploration to balance trade-offs without leveraging historical trends. We will incorporate these discussions in our revision.
>
> - **Our algorithm for BwK in both worlds:**
>   We introduce BwK in both worlds to demonstrate the versatility and broad applicability of our triple-optimistic framework. We will explicitly describe the problem setting and background to ensure clarity.
>
> - **Other typos and careless expression:**
>   We greatly appreciate your meticulous feedback and have carefully modified all typos in our revision. We have conducted a full manuscript audit to eliminate any residual oversights.
>
> We emphasize that our results **are unaffected** by the typos. We sincerely apologize for any confusion caused and deeply appreciate the reviewer's invaluable feedback. We hope the reviewer can re-evaluate our work in light of these clarifications.
>
> ---
>
> **Reference**
>
> [R1] Rakhlin, Sasha, and Karthik Sridharan. Optimization, learning, and games with predictable sequences. *Advances in Neural Information Processing Systems (NeurIPS)*, 2013.
>
> [R2] Mokhtari, Aryan, Asuman E. Ozdaglar, and Sarath Pattathil. Convergence rate of O(1/k) for optimistic gradient and extragradient methods in smooth convex-concave saddle point problems. *SIAM Journal on Optimization*.

---

> > ### Comment · Reviewer_MfNn · 2025-04-04
> >
> > I see! Along with the expectation drifting across the product, there was also a square that made its way into the expectation in Lemma 5.6, which I missed. Apologies for not reading carefully enough to catch this.
> >
> > I see that the results do hold, and will update my review accordingly. Thanks for the clarifications - I hope that the point made about $\\|g\_t(x\_t) - g\_{t-1}(x\_{t-1})\\|$ also makes it to the paper, since that offers a very clear explanation of the challenge in removing the knowledge of the context distribution, and also makes it evident precisely where it is needed in the analysis.

---

> > > ### Author Response · Authors · 2025-04-05
> > >
> > > We sincerely appreciate your positive feedback and insightful suggestions, which have been invaluable in helping us improve our work! We will incorporate a very detailed explanation about the challenge of removing knowledge to context distribution. Thank you once again for your time, expertise, and thoughtful review!

---

### Official Review · Reviewer_G6Cx · 2025-03-14

**Overall Recommendation:** 3

**Summary:**

The submission studies contextual bandits where the agent aims to minimize the regret (Eqn. (4)) and the constraint violation (Eqn. (5)) simultaneously. The submission shows that the previous bounds can be improved (in the sense of removing the need for Slater’s condition or lowering the upper bounds) by a clever choice of the dual update mechanism (i.e., the update of $Q_t$ in line 189R). The key inequality is Lemma 5.3 and the main result is Theorem 5.1. The proposed method, Optimistic3, is applied to solve CBwK, multi-armed bandits, and the best-of-both-worlds problem.


## Update after rebuttal

The authors' reply addressed my concerns. Thus, I will maintain my score.

**Claims And Evidence:**

Theoretically, the submission upper bounds the regret and the cost (Theorem 5.1). In the experiments, the performance curves show that the proposed method (Optimistic3) outperforms the existing baselines in terms of regret and cost (Figure 1).

However, there are several disconnections within the analysis and between the literature. Please see the comments/questions in the next two sections.

**Essential References Not Discussed:**

The related work discussed in Section 2 is fine.

**Experimental Designs Or Analyses:**

The experimental results show that the proposed method (Optimistic3) achieves lower regret and lower cost than the baselines. However, LOE2D shows a comparable performance. This is in contrast to the non-optimal upper bounds listed in Table 1 (the (Guo & Liu, 2024) row). Perhaps LOE2D is good enough, but poorly analyzed?

**Methods And Evaluation Criteria:**

What is the intuition of the 3rd optimistic update (line 192R)? A high-level discussion of its design and comparison can be found between lines 225L – 237L, but there is no further connection to the corresponding mathematical derivations.

**Other Comments Or Suggestions:**

The dual update is novel. The submission achieves better bounds. If the submission can refine its discussion of its contributions and the connections between the literature, it would provide a comprehensive knowledge to this line of research.

**Other Strengths And Weaknesses:**

A minor issue: the sentence in line 658 is misplaced.

**Questions For Authors:**

Please see the questions in the sections above.

**Relation To Broader Scientific Literature:**

Instead of focusing on regret minimization, the submission analyzes multiple measures for the bandit problem.

**Theoretical Claims:**

- How is the first equation (line 293R) derived from (4)? How did the previous work analyze regret under the primal-dual framework? We need a reference and an explanation to clarify which parts of the analysis follow the previous framework and which parts are novel.
- Lemma 5.3 provides a crucial inequality in the analysis. However, there is no explanation of its connection to the primal-dual framework in either the main text or the appendix.
- There is no discussion of several key concepts such as the pushback lemma and the function $h()$. This submission only writes down the derivation lines, but lacks the connection to the existing analysis in the literature and fails to articulate the novelty of the submission. For instance, besides the novel dual update rule (line 192R), is there any new technical tool developed from this submission? Is the pushback lemma (line 277L) novel on its own, or is it a refinement of an existing result?
- The pushback lemma is applied in lines 557 and 660. What is its intuition? What is pushback to what?
- The submission claims to be able to relax Slater's condition. In which line of the derivation can we see that the novel dual update removes the need for Slater's condition? Again, without referencing the literature and discussing the details of the derivation, it is difficult for a reader to understand the novelty of the submission.

---

> ### Author Rebuttal · Authors · 2025-04-01
>
> We sincerely thank the reviewer for the positive feedback. We would like to address your questions as follows.
>
> - **"Tripe-optimism".** The intuition behind our optimistic dual update stems from the Optimistic Online Mirror Descent in [R1], which accelerates convergence through effectively using predictions. Building on its success in min-max optimization [R2], we adapt this paradigm to design a momentum-driven dual update mechanism. Mathematically, our update incorporates a correction term that acts as a momentum term to predict and exploit trends in the reward-cost tradeoff dynamics, thereby accelerating convergence. We will incorporate these discussions in our revision.
>
> - **Regret analysis and prior works.** The regret in (4) measures the performance gap between the algorithm's policy and the optimal fixed policy in hindsight, where $\mathbb{E}[\langle \pi^* - \pi_t, \mathbf{f}(x_t) \rangle] $ represents the instantaneous regret at step $t$. Classical primal-dual methods (e.g., Slivkins et al., 2023) necessitate manual bounding of the dual variables and further decouple regret and constraint violation guarantees by independently bounding primal and dual regret terms. Recent work (Guo & Liu, 2024) introduced a refined Lyapunov drift framework to eliminate the knowledge requirement of Slater's condition but still require it hold. In contrast, our approach integrates a unified analytical framework via the pushback lemma (Lemma 5.3), enabling simultaneous control of regret and constraint violations. The optimistic design further facilitates bounding $Q_t$ by incorporating a momentum-driven correction term, circumventing the need for the Slater condition entirely. We will include a comprehensive comparative analysis of these methodologies.
>
> - **The pushback lemma.** Pushback lemma is a fundamental result in online optimization, particularly when working with Bregman divergences (here we specify as KL divergence). This lemma comes from the strong convexity property and guarantees that the optimization process inherently resists deviations from the optimal solution, with the term $\alpha D(\pi||\pi_{opt})$ quantifying the magnitude of this "pushback" effect.
> Here, $h(\cdot)$ denotes the primary objective function; it forms an OMD-style update as in (7) when combined with the regularization term. By specifying $h(\cdot)$ as the surrogate function in (6), the solution $\pi_{opt}$ corresponds directly to the policy $\pi_t$.
> We have included the discussion of these key terms and supplemented it with additional references to enhance clarity and accessibility.
>
> - **Comparable performance with LOE2D.** Although both Optimistic$^3$ and LOE2D significantly outperform LagrangeCBwLC (which requires knowledge of Slater's condition), there remains a noticeable gap between them, which appears smaller due to plotting averages. Unlike LOE2D, our approach employs an optimistic design for $Q_t$, which accelerates convergence to the optimal reward-cost tradeoff. This design enables a more refined theoretical analysis that cannot be achieved through LOE2D.
>
> ---
>
> **Reference**
>
> [R1] Rakhlin, Sasha, and Karthik Sridharan. Optimization, learning, and games with predictable sequences. Advances in Neural Information Processing Systems (NeurIPS), 2013.
>
> [R2] Mokhtari, Aryan, Asuman E. Ozdaglar, and Sarath Pattathil. Convergence rate of O(1/k) for optimistic gradient and extragradient methods in smooth convex-concave saddle point problems. SIAM Journal on Optimization.

---

> > ### Comment · Reviewer_G6Cx · 2025-04-08
> >
> > Thank you for your response. It addresses my concerns. I would like to keep my score.

---

> > > ### Author Response · Authors · 2025-04-09
> > >
> > > Thank you very much for your acknowledgment and the thoughtful feedback. We sincerely appreciate the time and effort you dedicated to reviewing our work!

---

### Decision · Program_Chairs · 2025-05-01

**Decision:**

Accept (poster)

**Comment:**

This paper studies contextual bandits with general constraints, where the learner must simultaneously minimize regret and control constraints violations. The authors proposed a triple-optimism framework, and theoretically show that the proposed approach achieves $\sqrt{T}$-type bounds on both regret and constraint violations, without knowing the Slater's condition. The authors have addressed most of the reviewers’ concerns during the rebuttal, and the majority of reviewers are now leaning toward acceptance.